



# Pleistocene climate characteristics in the most continental part of the northern hemisphere: insights from cryolithological features of the Batagay mega thaw slump in the Siberian Yana Highlands

Kseniia Ashastina [1], Lutz Schirrmeister [2], Margret Fuchs [3], Frank Kienast [1]

[1] Senckenberg Research Institute and Natural History Museum, Research Station of Quaternary Palaeontology, Weimar, 99423, Germany
[2] Alfred Wegener Institute Helmholtz Centre for Polar and Marine Research, Potsdam, 14471, Germany
[3] Helmholtz-Zentrum Dresden-Rossendorf, Helmholtz-Institute Freiberg for Resource Technology, Freiberg, 09599, Germany

*Correspondence to*: Kseniia Ashastina (Kseniia.Ashastina@senckenberg.de)

**Abstract.** Syngenetic permafrost deposits formed extensively on and around the arising Beringian subcontinent during the Late Pleistocene sea level low stands. Syngenetic deposition implies that all material, both mineral and organic, gets frozen parallel to sedimentation and remains frozen until degradation of the permafrost. Permafrost is therefore a unique archive of late Pleistocene paleoclimates. Most studied permafrost outcrops are situated in the coastal lowlands of NE Siberia and are

thus under certain influence of today's rather maritime climate. Permafrost sections more inland are in contrast scarcely available. Here we describe the cryolithological and geochronological characteristics of a permafrost sequence near Batagay in the Siberian Yana Highlands, the interior of the Republic Sakha (Yakutia), Russia. The recently formed Batagay mega thaw slump exposes permafrost deposits to a depth of up to 80 m and gives insight into a sought climate record close-by the Pole of Cold - the place with the most severe continental climate of the Northern Hemisphere. We provide a detailed

stratigraphic description of this profile and present results of cryolithological and geochemical analyses to deduce the genesis of the permafrost sequence, which comprised, according to our observations and sedimentological results, five lithological units. Geochronological dating (OSL and [14]C ages) and stratigraphic implications delivered a temporal frame from the Middle Pleistocene to the Holocene for our sedimentological interpretations and also revealed interruptions in the deposition of the sequence. The sequence of lithological units indicates a succession of several distinct climate phases: a middle

Pleistocene Ice Complex indicates cold stage climate conditions resulting in a mean annual ground temperature at least 8 °C lower than today; then, ice wedge growth stopped due to highly increased sedimentation rates and eventually a rise of temperature; full interglacial climate conditions existed during accumulation of an organic-rich layer - plant macrofossils reflected open forest vegetation existing under dry conditions during MIS 5e, the late Pleistocene YIC (MIS 4-2) proves again severe cold-stage climate conditions with a mean annual ground temperature 8 to 10 °C lower than today. In the

Holocene cover, no alas deposits indicating thermokarst processes, were detected. The main focus of our studies was material from the late Pleistocene Yedoma Ice Complex. The permafrost section was sampled over a depth of 60 m and analyzed for a range of sedimentological properties. The sequence is composed mainly of fine-sand with percentages from 40 % to 70 % varying between as well as within the units. Total organic carbon changes from 0.1 wt % to 4.8 wt %,



magnetic susceptibility values are within the range of 13.7-30 SL. A detailed comparison of the permafrost deposits exposed in the Batagay thaw slump with well-studied permafrost sequences, both coastal and inland, is made to highlight common features and differences in their formation processes and palaeoclimatic histories. Despite stratigraphical similarities to coastal outcrops, the Batagay sequence differs in some characteristics from them. Fluvial and lacustrine influence is common for certain depositional periods in the majority of permafrost exposures but have to be excluded for the Batagay sequence. We interpret the characteristics of Yedoma deposits at this location as a result of various involved climatically induced processes that are partly seasonally controlled: nival deposition might have been dominant during winter time, whereas proluvial and aeolian deposition could have prevailed during the snowmelt period and the dry summer season.

**Key words**: Beringia, Pleistocene, Batagay thaw slump, permafrost, Yedoma, Ice Complex

## 1 Introduction

During Pleistocene marine regression stages, ice-rich deposits several dozen meters in thickness - the Yedoma Ice Complex (YIC), formed on the now inundated Laptev and East Siberian Sea shelves and on the coastal lowlands of northern Yakutia (Romanovskii et al., 2000a; Schirrmeister et al., 2013). Because they contain syngenetically-frozen sediments and well-preserved fossil remains, YIC deposits provide a unique late Pleistocene palaeoenvironmental archive. Due to their importance as sinks of organic carbon and as palaeoenvironmental archives, Ice Complex deposits have been of great scientific interest for decades (e.g. Kaplina, 1981; Giterman et al., 1982; Kienast et al., 2005; Sher et al., 2005; Walter et al., 2006). Nevertheless, the main climate processes that resulted in Ice Complex formation are still not yet fully understood and remain a subject of controversy (Schirrmeister et al., 2013; Murton et al., 2015). The concept of a purely aeolian origin of the mostly silty and sandy, ice-rich deposits has become a widely accepted view in recent time (Zimov et al, 2012; Astakhov, 2014; Murton et al., 2015), but the assumption that loess covered the whole area during the late Pleistocene contradicts palaeontological data that indicate the existence of a diversity of habitats including aquatic and palustrine environments (Sher et al., 2003; Kienast et al., 2005), as well as cryolithological studies (Schirrmeister et al., 2011b). For this reason, the hypotheses of nival formation (Kunitsky, 1989), proluvial and slope genesis (Slagoda, 2004), alluvial (Rozenbaum, 1981) or polygenetic genesis (Konishchev, 1981; Sher 1997) are noteworthy.

YIC deposits in Yakutia are mainly accessible at natural outcrops along the seacoast or at river banks, primarily in the coastal lowlands; these areas are now under the certain influence of a maritime climate. All discussed processes of YIC formation are either related to climate (aeolian, nival processes) or to geomorphology (slope, alluvial deposition). To distinguish between aeolian and other processes in the resulting formation, the examination of YIC deposits in locations with climate and morphology differing from that in the northern coastal lowlands, i.e. more inland and in mountainous areas, might contribute to a better understanding of the formation genesis by comparing the lithological characteristics in different localities.



The Yana Highlands represent such a location because they form the benchmark for an inland climate north of the Arctic Circle. Verkhoyansk, located in the Yana Highlands, is recorded as place of the Pole of Cold; the Yana Highlands represent thus the region with the most severe climatic continentality in the Northern Hemisphere (Voeikov Main Geophysical Observatory, 1981; Harris et al., 2014). Kunitsky et al. (2013) reported on a rapidly proceeding permafrost thaw slump near

Batagay, Verkhoyansky district, Sakha Republic (Yakutia), which has grown tremendously in the past 30-40 years. Due to thermo-denudation rates of up to 15 m per year, the mega thaw slump reached a width of up to 800 m in 2014 (Günther et al., 2015). Situated in the Yana Highlands (Fig. 1), the Batagay exposure formed unaffected by fluvial or coastal abrasion processes. It is one of the few active permafrost outcrops in interior Yakutia that exposes a long climate record of the Late Pleistocene or even older age (Fig. 2).

Previous studies on the Batagay permafrost exposure reported on the structure and composition of the upper 12.5 m of the outcrop, and discussed thermal denudation processes (Kunitsky et al., 2013), estimated expansion rates using remote sensing data (Günther et al., 2015), or described findings of mammoth faunal remains, including carcasses of horses (*Equus* sp.) and bison (*Bison priscus*), as well as bone remains of cave lions (*Panthera leo spelaea*), woolly rhinoceroses (*Coelodonta antiquitatis*), mammoths (*Mammuthus primigenius*), and other extinct Pleistocene animals (Novgorodov et al., 2013).

In this study, we describe the structural and sedimentological characteristics of the Batagay permafrost sequence. The main aims of our study are (i) to deduce a cryostratigraphical classification of this exceptional YIC sequence and its underlying units in comparison to other YIC records in northeastern Siberia; (ii) to differentiate the depositional processes and underlying climate conditions; and (iii) to highlight common features of and differences between coastal and inland YIC sequences in Yakutia to shed light on their formation processes and palaeoclimate history.

**1.1 Study site**

The Batagay outcrop (67°34´41.83´´ N, 134°45´46.91´´ E) is located 10 km southeast of Batagay, the municipal center of the Verkhoyansk district, Sakha Republic (Yakutia). The study site is located on the left bank of the Batagay River, a tributary of the Yana River, and cuts down between 300 and 240 m asl into the foothills of Mount Khatyngakh, 381 m high (Fig. 1c). According to Günther et al. (2015), the height difference between the headwall and the outflow of the slump into the Batagay

River is 145 m along a distance of 2300 m, while the maximum slump width is 800 m.
The study area belongs to the western side of the Verkhoyansk-Kolyma Orogen, which is characterized by the occurrence of Tertiary dark grey terrigenous siltstone (alevrolits) and argillite, mudstone that has undergone low-grade metamorphism (Vdovina, 2002; Fig. 3 geological map). Both siltstone and mudstone deposits contain layers of sands forming crumpled and broken sediment packs with intrusive rocks. In places, a weathered clayey crust covers the Neogene rocks. The Neogene is

represented by clay deposits interspersed with pebbles and gravel, loam, sandy loam, and sands. Quaternary deposits are present as discontinuous layers covering older beds of hard rock and dispersed rocks (Kunitsky et al., 2013).
Meteorological observations are recorded at the Verkhoyansk weather station continuously since 1888 and revealed the globally greatest temperature gradient between summer and winter with a mean July temperature of $+15.5^0$ C and an



absolute maximum of +37.3 $^0$C as well as a mean January air temperature of - 44.7 $^0$C with an absolute minimum of -67.8 $^0$C. This is accepted as the lowest temperature measure on the Northern hemisphere. Verkhoyansk is therefore considered the northern Pole of Cold. The mean annual precipitation is only 181 mm with the lowest rate during the summer (56%), and the highest rate during the winter months (75%) (USSR Climate Digest, 1989).

The location of the study area in the coldest part of the Northern Hemisphere, is reflected by a Mean Annual Ground Temperature of -7.7 $^0$C (Romanovsky et al., 2010) and a permafrost thickness of 300-500 m (Yershov and Williams, 2004). The permafrost formation, which started during the Late Pliocene, was most likely influenced by local glaciers from the Chersky and Verkhoyansky mountains (Grinenko et al., 1998). Ice wedge casts in the Kutuyakh beds along the Krestovka River, northeastern Yakutia, indicate that permafrost existed in northern Yakutia already in the Late Pliocene (Kaplina,
10  1981).

Resembling deposits in the Yakutian coastal lowlands (Kaplina et al., 1980; Nikolskiy et al., 2010), thick YIC deposits also exist in the valleys of the Yana Highlands (Katasonov, 1954; Kunitsky et al., 2013) as well as along the Aldan River in central Yakutia (Markov, 1973; Pèwè et al., 1977; Baranova, 1979; Pèwè & Journaux, 1983). As the result of intense thermal degradation, the Batagay mega slump formed in just 40 years and cut about 60-80 m deep into ice-rich permafrost deposits
(Kunitsky et al., 2013), dissecting the Ice Complex down to the bedrock at a depth of 110 m below ground surface (m bgs) or 240 m above sea level (asl) (Vdovina, 2014, personal communication). A characteristic feature for the contact zone to the bedrock is the presence of cryogenic eluvium, frost weathering products of the siltstone that overlays leucogranite (alaskite).

## 2 Methods

We described the Batagay permafrost sequence during the June 2014 field campaign. We used a Nikon D300 SLR camera to
take photographs to be used for cryolithostratigraphical classifications. A Hama polarizing filter was used to highlight ground ice bodies for differentiating the cryolithological units. The 60-m-high outcrop was sampled from top to bottom along its height, ideally in one-meter steps, but depending on its accessibility. The profile was sampled along three different transects: Section A (0 to 10 m bgs), section B (40 to 50 m bgs), and section C (1 to 44 m bgs) (Figs. 4, 5). Since the steep outcrop wall was not approachable due to the danger of falling objects over most of its length, samples were taken mainly
from thermokarst mounds (baidzherakhs) in section C (Fig. 4b). The sampling procedure was as follows: The cryolithological characteristics at each sampling point were described and photographed, the sampling zone was cleaned, and frozen deposits were taken using a hammer and a chisel and placed into plastic bags. The wet sediments were air-dried and split into subsamples for sedimentological and biogeochemical analysis in the laboratories of the Alfred Wegener Institute in Potsdam.

Grain-size analyses of the < 2-mm fraction were carried out using an LS 200 Laser Particle Analyzer (Fa. Beckman-Coulter). Total carbon (TC) and total nitrogen (TN) were measured with a VARIO-EL-III Element Analyzer while the total organic carbon (TOC) content was measured with a VARIO-MAX Analyzer. Using the TOC and TN values, the TOC/TN (C/N) ratio was calculated to deduce the degree of organic matter decomposition. The lower the C/N ratio, the higher the



decomposition degree and vice versa (White, 2006; Carter and Gregorich, 2008). For TOC and stable carbon isotope ($\delta^{13}$C) analyses, samples were decalcified for 3 h at 95 °C by adding a surplus of 1.3 N HCl. Total inorganic carbon (TIC) content was calculated by subtracting TOC from TC. Using TIC values, the carbonate content as $CaCO_3$ was estimated via the ratios of molecular weight. The $\delta^{13}$C of TOC values were measured with a Finnigan DELTA S mass spectrometer and expressed in

delta per mil notation ($\delta$, ‰) relative to the Vienna Pee Dee Belemnite (VPDB) standard with an uncertainty of 0.15 ‰. Variations in $\delta^{13}$C values indicate changes in the local plant association and in the degree of organic matter decomposition (Hoefs and Hoefs, 1997). Lower $\delta^{13}$C values correspond to less-decomposed organic matter, while higher $\delta^{13}$C values reflect stronger decomposition (Gundelwein et al., 2007). Mass-specific magnetic susceptibility (MS) indicative of magnetic and magnetizable minerals was measured using Bartington MS2 instruments equipped with the MS2B sensor type. The data are

expressed in $10\text{-}8\ m^3\ kg^{-1}$ (SI).

For accelerator mass spectrometry (AMS) radiocarbon dating in Poznan Radiocarbon Laboratory, Poland, we picked out terrestrial plant remains. Possible reservoir effects as a result of the accidental use of freshwater aquatics are thus eliminated. The AMS Laboratory is equipped with the 1.5 SDH-Pelletron Model "Compact Carbon AMS" ser. No. 003 (Goslar et al., 2004). The results are presented in uncalibrated and calibrated $^{14}$C years. The calibration was made with the OxCal software

(Bronk Ramsey, 2009).

The lower part of the permafrost exposure was sampled for Optical Stimulated Luminescence (OSL) dating. Two samples were taken in form of cores from unfrozen but observably undisturbed deposits at the outer margin of thermokarst mounds. The tubes were sealed with opaque tape and transported to the OSL laboratory of TU Bergakademie Freiberg, Germany. One separate sediment sample was taken for HPGe high purity, low level gamma-spectrometry in order to determine the

radionuclide concentration required for dose rate calculations. OSL samples were treated under subdued red light. The outer 2 cm material layer was removed to retrieve only the inner core part that was not exposed to any light during sampling. The outer material was used for in-situ water content measurements. The inner core part was processed for quartz and feldspar. Quartz procedures yielded sufficient material in the 90-160 μm as well as in the 63-100 μm fractions, while K-rich Feldspar yielded only sufficient quantities for one sample in the 63-100 μm fraction. The chemical mineral separation and cleaning

included the removal of carbonates (HCl 10 %) and organics ($H_2O_2$ 30 %). The feldspar was separated from quartz using feldspar-flotation (HF 0.2 %, pH 2.4-2.7, dodecylamine). Subsequently, the density separation was performed to enrich K-feldspars (2.53-2.58 g/cm³) and quartz (2.62-2.67 g/cm³). Quartz extracts were etched (HF 40%) to remove the outer 10 μm of individual grains. After a final sieving, homogeneous sub-samples (aliquots) of quartz and K-feldspar extracts were prepared as mono-grain layer on aluminium discs within a 2 mm diameter. OSL and IRSL measurements were performed

using a TL/OSL Risø Reader DA-20 (Bøtter-Jensen et al. 2003) equipped with a 90 Sr beta irradiation source (4.95 Gy/min). Feldspar signal stimulation was performed at 870 nm with IR diodes (125 °C for 100 s) and the emission was collected through a 410 nm optical interference filter to cut of scattered light from stimulation and detected with a photomultiplier tube (Krbetschek et al. 1997). For quartz, blue LEDs of 470 nm were used for signal stimulation (125 °C for 100 s) and detection were done through a U 340 Hoya optical filter. Preheat and cut-heat temperatures were set to 240 °C and 200 °C



respectively. The measurement sequence followed the single-aliquot regenerative-dose (SAR) protocol according to Murray and Wintle (2000), including tests of dose recycling, recuperation and correction for sensitivity changes. Appropriate measurement conditions were evaluated and adjusted based on preheat and dose-recovery tests (Murray and Wintle 2003). Processing of measured data and statistical analyses were performed using the software Analyst v4.31.7 (Duller 2015) and

the R package 'Luminescence' for statistical computing (Kreutzer et al. 2012). Sets of 10-40 equivalent doses for individual samples and grain size fractions were analysed for skewness and data scatter. To address sediment mixing that potentially affects permafrost sediments, age modelling was based on the central age model (CAM, Galbraith et al. 1999).

## 3 Results

### 3.1 Field observations and sampling

Differences in the thawing rates along the outcrop provide a variety of conditions on the bottom and along the margins of the thaw slump. The western, northwestern and southwestern parts of the outcrop consist of vertical walls which are eroding most actively (Fig. 6), while the southeastern side of the thermo-erosional cirque is a gentler slope with a gradient of up to 45° (Fig. 4b). Along the western and southern parts of the outcrop, melt water and mud constantly flow off the steep slopes and form vertical drainage channels. The mud streams flowing downwards from the outcrop walls dissect a number of ≤30 m

high ridges of frozen sediments on the bottom of the thermo-erosional gully, forming a fan that is visible in the satellite photo (Fig. 4a). Due to a slight northeastern inclination, the sediment-loaded meltwaters stream down to the Batagay River.
The outcropping sequence is composed of five visually-distinct units with thickness changing along the outcrop (Fig. 6a). When the thickness of units is discussed, we refer to sections A and B unless otherwise stated (Fig. 6a-e, Table 1). Owing to the hillside situation of the outcrop, the position of the ground surface differs between sections A and C and, thus, the depth

bgs is only conditionally comparable between both sections.
Unit I represents the active layer with a thickness varying from the southeast to the northwest wall of the exposure between 1.4 m and 0.85 m, as measured at the end of June 2014. The well-bedded sandy sediments of Unit I were deposited in sublayers, 1-2 mm thick. The ≈9-cm-thick modern vegetation sod is underlain by a homogeneous, brown- to grey-coloured horizon containing numerous inclusions of charcoal and iron oxide impregnations (Fig. 7b). The upper part of the layer is

penetrated by modern roots. The lower boundary of Unit I is separated sharply from the underlying Unit II (Fig. 7a).
Unit II consists of YIC, 30-40 m thick, composed of silty-sandy sediments in a layered cryostructure enclosed by syngenetic ice wedges, 0.08-0.2 m thick in northwestern part and ≤6 m wide in western to southeastern part of the exposure (Figs. 6b, d). Unit II can be described, according to unaided eye observations, as follows. The northwestern part of the YIC can be divided into three subunits that mainly differ in their ice contents; this difference results in unequal resistance to thermal

erosion. Ice wedges are pronounced at the base of Unit II and gradually become less distinct upwards. This lower subunit of the YIC is the thickest of Unit II reaching 20-25 m here. Owing to less pronounced ice wedges and, as a result, increased thermal erosion, the middle subunit of Unit II is notched and forms a concave contour in the profile at the steepest point of





the outcrop (Fig. 6d). In contrast, the uppermost YIC subunit is stabilized by a massive ice wedge system resulting in a cliff overhang. The middle and upper subunits of unit II are each about 8 m thick. The southern part of unit II can also be visually divided into three subunits, differences in ice content are not obviously prominent, but the contour of profile reveals an upper and a lower strata, each 8 m thick, and the middle, 20 m thick subunit. The deposits are characterized by grey to brown mineral-rich horizons, which alternate with thin ice-rich layers, 0.2 to 7 cm thick (Fig. 7c). The YIC deposits contain more or less evenly distributed organic material, mainly in the form of plant detritus and the vertical roots of herbaceous plants. Occasionally, layers and patches with higher organic content can be found, e. g. a 0.2 m wide and 0.12 m thick brown fossil ground squirrel nest with a high amount of plant remains (Fig. 7d).

Unit III consists of frozen sediments that are rich in large macroscopic plant remains including numerous branches and twigs of woody plants. This horizon is detectable over the whole distance of the outcrop, mostly as a relatively thin layer with a sharp boundary below the base of the YIC (Figs. 6a, e). In several places, however, there exist accumulations of Unit III organic matter ≈5m thick filling former depressions that resemble ice wedge casts or small thermo-erosional drain channels (Fig. 7e). Unit III was sampled in the lower part of such a pocket-like accumulation below the coarse woody layer at a depth of about 40 to 42 m bgs. The samples taken in section B consist of organic material including numerous seeds, fruits, and plant debris in a distorted fine bedding alternating with silty fine sand beds (Fig. 7f).

Unit IV, which reaches a thickness of ≈25 m, approximates to the bottom of the exposure in most places. Unit IV is composed of distinct horizontally-layered frozen sediments (Fig. 6a, 6e) that are traceable without interruption over large distances along the steepest part of the outcrop (Fig. 6b). Unit IV is separated sharply from the overlying Unit III (Fig. 6e). In contrast to the YIC, Unit IV is neither penetrated by thick ice wedges, nor does it contain regular ice-wedge casts. Its cryostructure is dominantly horizontal, with sediment beds 5-20 cm thick separated by ice belts. Exposed exclusively at the steepest part of the profile, Unit IV was not accessible for sampling due to the danger of objects frequently falling from the >60 m high, intensely thawing and eroding, partly overhanging permafrost wall.

Unit V is exposed only at the deepest part of the thaw slump near the bottom of the profile (Fig. 6). The main part of this unit is not outcropping but buried. Even though only the truncated heads of ice wedges were exposed, the general composition of Unit V was easily observable and revealed ice-rich deposits in a layered cryostructure similar to the deposits of Unit II, embedded in syngenetic ice wedges ≤4m wide (Fig. 7g). Since Unit V exhibits distinct, separate ice wedges several meters wide beneath the layered Unit IV, it can be assumed to be a second Ice Complex older than the YIC. Unfortunately, Unit V was also not accessible for sampling.

**3.2 Geochronological results**

A total of nine radiocarbon dates are available covering ages from modern to non-finite (Table 2). For the uppermost Unit I, one [14]C AMS date of 295 years BP is available from a sample taken directly above the permafrost table indicating rather recent deposition of the active-layer sediments. In Unit II, seven samples were radiocarbon-dated. Material from 2.05 m bgs in section A resulted in a date of 33 ± 0.5 ka BP, while plant material collected from a ground squirrel nest at 4.6 m bgs in





section A (Fig. 7c) revealed a $^{14}$C AMS date of 26 ± 0.22 ka BP. In section C, dating results from 12.5 m and 14.5 m bgs present non-finite ages of > 48 ka BP and >51 ka BP, whereas sediment material from 18.5 m bgs was dated to 46 ± 2 ka BP. In section C, we collected organic material with very well-preserved plant remains embedded in frozen ice-rich permafrost sediments. We assumed in situ preservation of old material in excellent condition. Dating of this sample, taken at a depth of

24.5 m bgs, revealed, however, that this material is of modern (1991 – 2005 AD) origin and was most likely eroded from the top and later refrozen in the wall. Unit III deposits are situated directly below the YIC. One sample was taken for $^{14}$C AMS dating from the lower part of a sediment-filled depression about 6 m below Unit II in section B at a depth of 44 m bgs. The dating resulted in an infinite age of >44 ka BP.

OSL measurements for section B show that luminescence signals of quartz already reach the saturation level. For the two

duplicate samples in 47 m (sample 2.7/B/1/47 and 2.7/B/2/47) and the one in 50 m bgs (sample 2.7/A/2/50) only 11-26 out of 20-40 measured aliquots yielded equivalent doses and met the quality criteria of a recycling ratio within 10% and a recuperation of below 5%. Because of no significant skewness (below 1.5), age modeling was based on the central age model (CAM) according to Galbraith et al. (1999). However, the determined equivalent doses for several aliquots were still above the linear range of growth curves indicated by values above 2 times the D0 value and also by underestimation of

applied doses during dose recovery tests. Hence, for the two measured grain sizes of the three samples, only minimum ages could be determined (see Table 3). Only for the sample 2.7/B/2/47 in the grain size 63-100 µm an OSL age of 142.8 ± 25.3 ka could be calculated. A note of caution concerns the water content. OSL ages were based on in situ water contents, for this sample 34.3%, but samples were taken from unfrozen sediments, while the paleo-water content of the frozen section remains unknown. To give an upper boundary condition, the saturation water content was used as well, and then the age of this

sample yielded 160.9 ± 27.7 ka. Both age estimates lie at the common dating limits of OSL quartz techniques. For the same sample 2.7/B/2/47, also feldspar was available for luminescence dating. The feldspar grains of 63-100 µm showed bright IRSL signals and all 25 aliquots met the quality criteria. Determined equivalent doses were within the linear part of the growth curves and showed low errors and an extremely small data scatter resulting in low over dispersion values of 3.9% and no significant skewness (-0.32). The CAM yielded an IRSL age for feldspar grains of 210.0 ± 23.0 ka. Regarding the

saturation water content as an upper boundary condition of the paleo-water content, the IRSL age would increase by about 26 ka (see Table 3 and respective notes).

### 3.3 Sedimentological results

Analytical results are summarized for sections A and B in Figures 8, 9 and for section C in Figures 10, 11.

Unit I, representing the modern active layer, is composed of 44-59% fine sand with a mean grain size varying between 80 and 90 µm. The MS values are between 19 and 32 SI. The carbonate content is between 2.1 and 2.7 wt %. The TOC of the active layer was below the detection limit of 0.1 wt % in section A but about 1 wt % in section C. The TN values are about 0.12 wt %. Because the TOC content was insufficient at <0.1 wt%, δ13CTOC was not measurable and the C/N ratio could not be calculated.



Within the YIC, the mean grain-size of Unit II varies between 65 and 126 µm and is thus dominated by fine-grained sand. At about 30 m bgs, a distinct layer of medium-grained sand (mean diameter 253 µm) was detected. The MS values vary between 16 and 23 SI except for some higher values of 40, 31, and 43 SI at 43.5, 32.5, and 32 m bgs, respectively. The TOC ranges from <0.1 wt % to 4.8 wt %; higher values of ≥1 wt % were measured between 27.5 and 17.5 m bgs in section C and 7.4 and 4.6 m bgs in section A. The TN values range between < 0.1 and 0.49 wt %, while low TN values < 0.1 wt % are mostly accompanied by low TOC values. The C/N ratios are mostly low and range from 2.4 to 9.8. Only one sample at a depth of 32.5 m bgs shows a higher ratio of 13.1. The $\delta^{13}C$ values are rather uniformly distributed, ranging from -26.6 to -23.9 ‰ without any clear trend. The carbonate content is not stable within the profile and varies from 1.2 to 5.9 wt %, aside from one sample at 20.5 m bgs with a lower carbonate content of 0.03 wt%. Comparing the fine-grained sand fraction data and TOC contents, Unit II in section C could be subdivided into three subunits (Figs. 8, 9). The lower part of Unit II between 43.5 and 34.5 m bgs (Unit IIa) is dominated by fine-grained sand (>50 %) with low TOC (<0.1-0.7 wt %), whereas the middle part between 32.5-16.5 m bgs (Unit IIb) contains less fine-grained sand (20-50 %) and a higher TOC (0.7-4.8 wt %). The upper subunit at a depth from 16.5-8.5 m bgs (Unit IIc) is again mainly composed of fine-grained sand with low TOC.

The sedimentological characteristics of the lowermost part of Unit III were studied in section B with two samples from depths of 43 and 44 m bgs (Figs. 7f, 10, 11). The major fraction in the grain size distribution (GSD) of Unit III is fine-grained sand, accounting for 41-45 %. MS equals 30 SI. The TOC values are ≈3.3 wt %, the C/N ratio is ≈13, the $\delta^{13}C$ values are -26.5 to -26.1 ‰, and the carbonate content is 2.5-2.8 wt %. The sediment characteristics of Unit III are comparable to the ice- and organic-rich layers in Unit IIb.

Due to scarce accessibility, only one sample from Unit IV could be taken at 50 m bgs. According to the sedimentological characteristics of this material, Unit IV clearly differs from the overlying Units I to III. This sample is characterized by the largest sand fraction (70 %) and the highest carbonate content (8.2 wt %) of the studied sample set as well as the lowest MS value (13.7 SI).

## 4 Discussion

### 4.1 Lithostratigraphy

According to field observations as well as to geochronological and sedimentological data, the permafrost sequence of the Batagay mega slump consists of five distinct stratigraphic units (Fig. 6a). No gradual transitions were observed between the units, so erosional events or strong changes of accumulation conditions can be expected to have occurred.

Unit I represents the active layer or, as we call it, the Holocene cover. The presence of a Holocene layer is typical of the majority of permafrost exposures, although it differs in thickness and age; e.g. at Cape Mamontov Klyk it is 3 m thick and covers the time span from 9.5 to 2.2 ka (Schirrmeister et al., 2011b). The dating result from 1.15 m bgs yielded an age of 0.295 ka, which suggests that much of the Holocene layer was eroded. The thickness is not constant along the Batagay outcrop and reaches a maximum observed depth of 1.4 m.



Unit II corresponds to the YIC. YIC deposits can form only under extremely cold winter conditions. They are thus indicative of cold stage climate in a continental setting. Our dating results confirm the assumption that the YIC was deposited from at least >51 ka BP to 12 ka BP, thus during the last cold stage and including the MIS 3 (Kargin) interstadial period. Huge syngenetic ice wedges and high segregation ice contents are the most typical features of Yedoma sequences. The structure of ice wedges intersecting sediment columns are evidence for the syngenetic freezing of the ice-wedge polygon deposits. The ice wedges were 4.5 to 6.5 m wide, which indicates the impact of an extremely cold climate during their formation and also indicates aridity (Kudryavtseva, 1978). The thermokarst mounds (baidzherakhs) appearing in staggered order 4.5-6.5 m apart on the upper southeastern part of the YIC support this hypothesis.

The structural differences of the Unit II ice wedges suggest that they represent three generations of past ice-wedge growth. Also the threefold division of Unit II, as visible in its contour in the profile and in grain-size parameters and TOC content in section C, may reflect three different climate stages, e.g. MIS 4, 3, and 2 during YIC formation. In this case, the MIS 4 and MIS 2 cold stadial phases were characterized by relatively uniform landscape conditions with fine sand accumulation and low bioproductivity whereas the MIS 3 interstadial was characterized by changing accumulation conditions and higher bioproductivity. Unfortunately, the geochronological data do not support such subdivision since most dates are beyond the limit of the radiocarbon method. Also, we could not take samples directly from the visually different subunits in the western part near section A (Fig. 6d) to verify if sedimentological characteristics confirm the apparent visual differences. The YIC at section A differed from YIC in other parts of the exposure in having considerably smaller ice wedges outcropping. We considered the absence of visible large ice-wedges due to exposed intra-polygonal sediment sequences concealing the ice wedges at this place. Owing to the lack of large exposed ice wedges, this part of the sequence was, however, separated from the YIC and regarded as own unit by Murton et al. (2016).

Dating results may indicate that parts of the YIC could have been eroded. Taken at a depth of 2.05 m bgs, the uppermost dated sample of Unit II in section A has an age of ca. 33 ka BP. The dating of the next overlying sample with a position in Unit I only about 1 m above resulted in an age of ca. 0.3 ka BP. No Holocene sediments older than the 0.3 ka BP sample at 1.15 m bgs in section A overlying the YIC have yet been found in the Batagay mega slump, but this could be due to the difficulty of accessing the upper parts of the profile. The youngest YIC age in section A of about 26.2 ka BP originates from plant material collected in a ground squirrel nest 4.6 m bgs. The age inversion between 2.05 and 4.6 m might be the result of younger material actively transported by ground squirrels deep into their subterranean burrows for food storage.

The youngest YIC age from the Batagay thaw slump of about 12.7 ka BP was determined in section C (southeastern part) at 8.5 m bgs. This result stresses the difference between southeastern and northwestern parts of the outcrop. An age gap of several tens of thousands of years could be expected between the infinite age of >48 ka BP at 12.5 m bgs and the 12.7 ka at 8.5 m in section C. It is implausible that only four meters of YIC deposits were formed during more than 35 ka.

The observed stratigraphic hiatus of up to 12 ka atop the YIC was likely caused by post-depositional erosional events, such as widespread thermo-denudation or local thermal erosion of Early Holocene deposits. A sudden shift from deposition to erosion in consequence of intense warming during the late glacial - early Holocene (Bølling-Allerød) transition is a





characteristic feature of many Yedoma sequences in Yakutia (e.g. Wetterich et al., 2014; Schirrmeister et al., 2011b) and can also be readily assumed for the Batagay thaw slump. The uppermost boundary of YIC sequences as dated with the AMS radiocarbon method differs between 28 ka BP on the New Siberian Islands and 17-13 ka BP at various other sites. Available radiocarbon dates from mature alas depressions in central Yakutia reported on an age of 12 ka BP (Katasonov, 1979;

Kostyukevich, 1993).

The organic layer of Unit III below the base of the YIC (Unit II) is characterized by a high abundance of macroscopic plant material including woody remains. Plant macrofossil analyses detected numerous taxa characteristic of northern taiga forests as they occur today at the study site. The main components of the reconstructed vegetation were larch (*Larix gmelinii*) as well as birch (*Betula* spp.) and shrub alder (*Alnus fruticosa*), which occurred together with some indicators of open habitats

(Ashastina et al., 2015). The palaeobotanical results clearly indicate climate conditions during the formation of this layer. High values of TOC and C/N and low $\delta^{13}$C values reflecting increased bioproductivity and moderate organic-matter decomposition confirm this suggestion. These proxy records together with the position of Unit III below the base of the YIC, the infinite AMS date of > 44 ka BP of the sample and the OSL quartz date of 142.8 ± 25.3 ka of the sample taken from unit IV, indicate that Unit III probably formed during the MIS 5e interglacial (Kazantsevo). This assumption is in a good

agreement with data from the Lake El´gygytgyn (Tarasov et al., 2013), where the Eemian interglacial warmest period was from 127 to 123 ka. The organic layer of the Batagay Unit III is continuous throughout the outcrop and shows a uniform thickness of about 1 m, reaching up to 3.5 m thickness in putative ice wedge casts. Such a distribution might indicate the presence of a continuous paleosol.

The uniformly-occurring Unit IV with its characteristic horizontal bedding was observed over large distances along the

lower and very steep segment of the exposure wall. The lack of thick ice wedges or ice wedge casts indicates that the climate conditions during deposition of Unit IV were inappropriate for the formation of an Ice Complex directly below the last interglacial Unit III. Unit IV instead represents sediments that, in contrast to Ice Complex deposits, consistently accumulated under uniform depositional environments. We did not find any evidence for the presence of lacustrine or fluvial deposition in the sediments along the whole permafrost sequence. We detected neither pebbles nor other coarse material, nor freshwater

mollusk remains. Fluvial or lacustrine deposition can be excluded because of the topographical setting: The area around the Batagay mega slump is inclined northeastwardly. This would prevent water stagnation and would not result in clear horizontally-layered structures. Instead, laminar slope deposition as the result of ablation or aeolian activity can be assumed to be the main sedimentation processes that formed Unit IV. The assumed laminar slope deposition can be related to cryoplanation and other nivation processes during cold phases, with perennial snow accumulations further uphill. Since not

accessible for sampling during our field stay, detailed sedimentological results are not available for Unit IV. A detailed description of this unit was presented by Murton et al. (2016).

The lowermost Unit V was observed in the field at the bottom part of the thaw slump wall (Figs. 6, 7g). The existence of truncated ice wedges several meters in width and their position more than 20 m below Unit III, which represents the last interglacial period, allow the interpretation that this unit represents an Ice Complex indicating continental cold-stage climate



with extremely cold winters occurring already during the mid-Pleistocene. The contained symmetric ice wedges point to syngenetic formation of Unit V. The finding of such ancient ice wedges demonstrates also that ice-rich permafrost survived several glacial-interglacial cycles. Similar observations of Ice Complex deposits older than the last interglacial were made on Bol'shoy Lyakhovsky Island by Andreev et al. (2004) and Tumskoy (2012) and were dated by Schirrmeister et al. (2002).

An overview of changes in paleoclimatic conditions and the response to these changes reflected in the sediment sequence of the Batagay mega thaw slump is available in Table 4. The shifts in sedimentation characteristics of the Batagay sequence are in a good agreement with global climatic events, such as glacial and interglacial phases, recorded by oxygen isotope data, and regional climatic changes, identified by stadial/interstadial phases in Siberia and Europe.

## 4.2 Sedimentation processes of the Batagay YIC

Our reconstruction of YIC formation based on the analysis of GSD as discussed in the results´ section. Additional studies on the mineralogical composition as well as micromorphological analysis would be useful to identify the sources more precisely. The radiocarbon dating results of the YIC in the Batagay mega slump from >51 ka to 12 ka BP with large gaps in between suggest that the sedimentation experienced interruptions. Beside post-depositional erosion, the gaps within Unit II

might also be the result of temporarily- and spatially-shifted local deposition. Sediments were deposited during given periods and at a particular part of today's outcrop mainly from a certain source area, such as Mt. Kirgillyakh northwest of the outcrop; during earlier or later periods, sedimentation might have stopped there and, instead, have taken place mainly at another part of the foothill and from a different local source area, e.g. Mt. Khatyngnakh southwest of the outcrop (Fig. 1c). Due to varying discharge directions, locally restricted denudation phases might have also occurred. As a result, the entire

YIC sequence might not have formed simultaneously, but may have formed piecewise and successively.

We assume that the sediment material was subaerially exposed and was incorporated into the permafrost syngenetically, e.g. at the same time as the deposition. The final accumulation occurred within small depressions of low-center polygons which existed between the ice wedges. The exposed YIC wall is a cross section through the former landscape with polygonal patterned ground.

YIC subunits IIa and IIc are characterized by a unimodal distribution curve made up >50% by a fine-grained sand fraction; this can be explained as a result of periglacial, proluvial, or nival processes. We suggest that subunit IIc correlates to the MIS 2 (Sartan) stadial and IIa correlates to the MIS 4 (Zyryan) stadial. After 60 ka BP, local mountain glaciers no longer reached the highlands (as was true during the Middle Pleistocene), but glaciation covered only the western and southwestern Verkhoyansky Mountains (Siegert et al., 2007). The area around Batagay was, therefore, not glaciated during at least the last

60 ka; the bedrock in the study area could have been affected by strong frost weathering. Twenty km south of the Batagay thaw slump, a possible sediment supplier is located: Mat'- Gora, a 1622-m-high massif (Fig. 1b). We suggest however that Mt. Kirgillyakh and Mt. Khatyngnakh, situated just 2 km away, mainly provided substantial input to the sediment composition of the Batagay deposits.





According to Kunitskiy et al. (2013), nival processes were highly significant here during the late Pleistocene. They proposed that nival (snow-filled) depressions existed at this time, so that cryohydro-weathering, as discussed by Konishchev (1981), took place. The material trapped on top of the snow was incorporated into downslope sediments.

In addition to the nival genesis of the sediments, the material trapped by snow could have been transported there by local aeolian processes, as the coarse silt fraction of 30-50 μm suggests. Some horizons are characterized by less than 40% of silt in the GSD, thus indicating that aeolian input, although it is significant, might not have been the main deposition process. The Batagay mega slump is located within 10 km of the Yana River and 30 km from the Adycha River floodplains. The meandering pattern of both river systems and the adjacent sandy terraces ≤50 m high (Fig. 1c, upper left) suggest that during late Pleistocene the wide, braided floodplains could have provided material for local aeolian input. Local aeolian input could originate from the Batagay river floodplain as well (Murton et al., 2016). Even though the substrate is almost everywhere stabilized by vegetation, the sandy terraces of the Yana River also nowadays provide high amounts of material available for local dust storms in summer. The results of MS measurements did not display, however, any changes in the content of magnetic or magnetizable minerals within the studied sequence as would be expected from shifts of the main source areas, e.g. from local slope deposits to more regional, redeposited alluvial material from the Yana River.

The GSD curves for Units IIb and III indicate a polygenetic sediment origin; this is indicated by the bimodal distribution in fraction sizes, from silt and coarse silt - a possible aeolian transport indicator - to sand, a possible hint of proluvial and nival genesis, as was discussed for subunits IIa and IIc. Nevertheless, the high percentage of the silt fraction in the GSD of subunit IIb cannot be interpreted as an exclusive indicator of aeolian deposition, because high silt content in the sediment composition can also result from cryogenic disintegration of quartz due to repeated thawing and freezing cycles (Konishchev and Rogov, 1993; Schwamborn et al., 2012).

Seasonally-controlled processes under the influence of a continental climate might have governed the deposition of Unit II; during the cold winter, nival deposition could have been dominant, whereas proluvial and aeolian deposition could have prevailed during the snowmelt period and the dry summer season. Aeolian deposition was thus locally restricted and was one of several processes that formed the Batagay Ice Complex sequence.

### 4.3 Climatic implications in comparison with other Ice Complex sequences (inland versus coastal Ice Complex)

The studied Batagay mega slump shows a general structure comparable to coastal permafrost exposures of Quaternary deposits in northeastern Siberia, as described by Schirrmeister (2011a) as follows: (I) Late Saalian (MIS 6) ice-rich deposits (ancient Ice Complex), (II) Pre-Eemian floodplain deposits, (III) Eemian (MIS 5e) thermokarst deposits, (IV) Early-Last Glacial Period (MIS 5a) alluvial deposits, (V) Early (MIS 4), Middle (MIS 3), and late Weichselian (MIS 2) ice-rich deposits (YIC), and (VI) Lateglacial and Holocene (MIS 1) thermokarst deposits.

Using the abovementioned general structure of permafrost sequences in the coastal lowlands, we subsequently compare the Batagay units with other Quaternary sediment records in northeastern Siberia.



Unit V of the Batagay outcrop is represented by the heads of thick ice wedges indicating Ice Complex deposits older than the MIS 5e. Similar structures with a comparable stratigraphic position, corresponding to part I of the general structure of coastal permafrost exposures, were observed on Bol'shoy Lyakhovsky Island and dated back to 200 ka BP (Schirrmeister et al., 2002; Andreev et al., 2004; Tumskoy 2012). Ice Complex is syngenetically frozen sediment containing a grid-like system of

large ice wedges resulting in a ground surface pattern of polygonal ridges encircling small depressions, which during Ice Complex genesis, act as sediment traps. Polygonal ice wedges form due to repeated thermal contraction of the frozen ground at temperatures below a certain mean annual air temperature threshold (e.g. -8 °C, Plug and Werner, 2008) resulting in netlike arranged cracks that are filled in spring by snowmelt water, which immediately freezes and forms ice veins. Ice-wedge growth is not only influenced by climate but also by local factors such as ice content, grain size distribution,

vegetation and snow depth. Ice Complex characteristics such as spacing and width of ice wedges cannot reliably be used to estimate mean annual or even mean winter palaeotemperatures (Kaplina, 1981; Plug and Werner, 2008). However, Ice Complex deposits clearly indicate climate conditions much colder than present. According to Kaplina (1981), the mean annual ground temperature during Ice Complex aggradation was -20 to -25 °C, at least for the late Pleistocene (today -7.7 °C). Similar values can be assumed for the time of formation of the mid-Pleistocene Ice Complex - Unit V in the Batagay

profile.

Pre-Eemian deposits (as described for part II of the general classification) were detected on Bol´shoy Lyakhovsky Island (Schirrmeister et al., 2011b) and Oyogos Yar (Kienast et al., 2011). The position of Unit IV in the Batagay outcrop stratigraphically matches the abovementioned pre-Eemian floodplain deposits in the general classification of coastal permafrost exposures from Schirrmeister et al. (2011a). Unfortunately, we could not analyze material from Unit IV to

reconstruct its genesis, but according to our field observations (color and structure of the unit), it is unlikely that the material is of subaquatic origin. The main reason for the absence of temporary and permanent water bodies at the site might be the relief gradient and associated rapid drainage of surplus waters after snowmelt and permafrost thawing. In this setting, intensified rates of frost weathering of the surrounding mountains' bedrocks and increased slope deposition of alluvial material are regarded as the main deposition sources. Ice wedges or ice wedge casts as prevailing in the underlying Unit V

are absent in Unit IV. The abrupt transition between both strata suggests a cessation of ice complex formation owing to a sudden climate shift. We assume that ice wedge growth ceased because of boosted sedimentation disrupting frost cracks Also milder winter temperatures preventing thermal contraction and frost cracking are conceivable.

Unit III in the Batagay outcrop might be equivalent to part III of the general permafrost sequence structure with few differences. Its structure is referred to as a lake-thermokarst complex (Tomirdiaro, 1982) or as ancient Achchagyisky and

Krest Yuryakhsky alas deposits (Kaplina, 2011) and is displayed in peat layers ≤10 m thick filling former ground depressions, e.g. ice wedge casts. This horizon formed as result of permafrost thaw processes during the last interglacial (MIS 5e) warming and is present with variable thicknesses in all permafrost exposures from the coastal zone, e.g. from Duvanny Yar, Kolyma River (Kaplina, 1978) and Mus-Khaya, Yana River (Katasonov, 1954; Kondratjeva, 1974) to exposures further inland: e.g. the Allaikha outcrop and Sypnoy Yar, Indigirka River (Lavrushin, 1962; Kaplina and Sher,



1977; Tomirdiaro, 1983), and Mamontova Gora, Aldan River (Pewe, 1977). The peat horizon can occur uninterrupted or only in scattered peat lenses as is the case in the Allaikha outcrop. At the Batagay profile, we noticed a rather thin (about 1 m thick) layer with pronounced lenses ≤3.5 m thick filling former ground depressions. Gubin (1999) and Zanina (2006) studied the palaeosols of the ancient alas complex at Duvanny Yar and suggested that two types of soil occurred there: Peat bog soils

and peaty floodplain soils both indicating wet ground conditions. Palaeobotanical analyses of Unit III deposits at the Batagay profile revealed only terrestrial plant remains, no aquatic or wetland plants. Our data therefore suggest that, instead of prevalent bogs, northern taiga with dry open-ground vegetation existed at Batagay during the MIS 5e Interglacial (Ashastina et al., 2015). Limited peat accumulation and dry ground conditions during formation of Unit III might also be due to the relatively low ice content of the underlying Unit IV, which, when the MIS 5e warming started, resulted in less available melt

water from thawing permafrost.

The YIC, corresponding to Unit II in the Batagay profile, is the most-accessible and best-studied Quarternary permafrost sediment type in Siberia. As pointed out for the older ice complex in Unit V, polygonal ice wedge systems are indicative for continental cold stage climate with very cold winter temperatures and annual ground temperatures. The YIC developed during MIS 4 – MIS 2, Romanovskii et al. (2000b), indicated a mean annual ground temperature by 8 °C lower than today

during MIS 4 and MIS 3 and by 10 °C lower than today during MIS 2 for the coastal lowlands. Siegert et al. (2009) summarized the results of the Russian-German decadal cooperation on the investigation of coastal YIC in northeastern Russia with special attention to sites at Cape Mamotovy Klyk, the Lena Delta, Bykovsky Peninsula, Bol´shoy Lyakhovsky Island, and the northern islands of the New Siberian Archipelago. According to the dating results, coastal Ice Complexes were preserved until 27 ka BP at the New Siberian Archipelago, while along the mainland Laptev Sea coast, also younger

deposits are available. Konishchev (2013) suggested a YIC formation time frame from 50-11 ka BP. The youngest dates of Batagay YIC deposits have an age of 12 ka BP. A hiatus in the sedimentation record from 12 to 0.3 ka BP occurs in the Batagay sequence. Such gaps also exist in exposures at Kurungnakh Island, Lena Delta (Wetterich, 2008) and Molotkovsky Kamen, Malyj Anjuy River (Tomirdiaro and Chernenky, 1987); according to Kaplina (1981), sedimentation gaps are possibly connected to an increase in humidity and forest cover. The moister Holocene climate of this area was governed by

changes in the hydrological regime, which were triggered by the transgression of the Laptev and East Siberian Seas. Gaps in sediment preservation might also be explained by locally-increased erosion.

According to the general scheme of landscape types introduced in Schirrmeister et al. (2011b), the Batagay YIC is related to the second landscape type, which represents cryoplanation terraces occurring on foothill slopes. (The first landscape type is low-elevation coastal mountains and foreland accumulation plains; the third landscape type is extended lowland at a great

distance from mountain ranges.) Because the area around Batagay was not glaciated during at least the last 60 ka (Siegert et al., 2007), the bedrock could have been affected by intense frost weathering during the late Pleistocene providing fine-grained material for YIC formation. Such bedrock weathering is also typical of the permafrost sequences at Bol´shoy Lyakhovsky Island, Cape Svyatoy Nos, and the Stolbovoy and Kotel´ny islands (Siegert et al., 2009).



Based on detailed studies of YIC in Siberia, Katasonov (1954) detected a cyclic structure of sediments in the Mus-Khaya outcrop. This concept was further developed by Lavrushin (1963) and Romanovskii (1993) and summarized by Konishchev (2013). The identified lithogenetic cycles depict changes in climate conditions that occurred from MIS 4 to MIS 2, e.g. two stadial stages (Zyryan, Sartan) and one interstadial stage (Kargin) with several thermochrones within. On the basis of

sedimentological and TOC analyses, we also distinguished three subhorizons in the YIC structure (subunits IIa to IIc). According to Konishchev (2013), sediment sequences consisting of heavily-deformed greenish-gray ice-rich loam alternating with peat inclusions and less ice-rich, non-deformed strata of sediment and brown loam with a fine layered cryostructure are evident along the permafrost exposures, especially in the lower parts. Such sedimentation cycles, governed by floodplain setting, are mentioned e.g. for the Mus-Khaya, Duvanny Yar, and Chukochiy Yar outcrops (Kondratjeva,

1974; Kaplina, 1978; Konishchev, 2013). In Batagay, such cyclicity could not be detected partly due to the specific cross section of the steep southwestern permafrost wall (this wall was mostly cut along the thick ice wedges), partly owing to the lack of accessibility of the unit II along the whole exposure. But at the more gentle southeastern part of the outcrop, such structures also did not occur, possibly due to a different geomorphological setting (discussed below).

The subdivision of the Batagay YIC is similar to that of Mamontovy Khayata at the Bykovsky Peninsula (Schirrmeister et

al., 2011b). Accordingly, the middle parts of both YIC sequences contain MIS 3 peat horizons. In Unit IIb of the Batagay sequence, two horizons rich in organic carbon were identified. The upper part of Mamontovy Khayata is composed of proluvial MIS 2 (Sartan) deposits resembling Unit IIa in Batagay. However, the Batagay source material certainly differs from that in the coastal outcrops. Bykovsky was fed by the Khara-Ulakh Mountains, a low-elevation coastal mountain ridge; in contrast, Batagay was supplied with sediments from the hillside of the Kirgillyakh-Khatyngnakh eminence. Another

possible material source for Batagay is windblown material from the Yana and Adycha River valleys; this is suggested by the occurrence of sandy terraces adjacent to the Yana floodplain 7 km west of the Batagay outcrop (Fig. 1c, upper left part).

A certain proportion of local aeolian deposition in the formation of the Batagay YIC is indicated by its sedimentological characteristics. Despite similarities in the general YIC (Unit II) structure, the Batagay sequence is distinct from other permafrost exposures. All coastal outcrops are characterized by polymodal grain size curves, a dominance of fine-grained

sediments, and relatively high concentrations of silt in their structure. The Batagay YIC, in contrast, is dominated by fine-grained sand in a unimodal GSD curve (Units IIa and c) and by bimodal coarse-silt and fine-sand curves (Unit IIb). Higher concentrations of sand in the YIC exposures of Kurungnakh Island and Diring Yuriakh (Lena River Delta) are interpreted to be of aeolian origin (Siegert et al., 2009; Waters et al., 1997).

The characteristics of the Batagay YIC profile could be assumed to be close to the Mus-Khaya or Mamontova Gora

outcrops, because the first is located along the Yana River bank, and is in a comparable hydrological situation, while the second, from the Aldan River in Central Yakutia, is another example of an inland YIC that never experienced maritime influence. Although also situated in the catchment area of the Yana-Adycha River system, the Mus-Khaya Ice Complex (Katasonov, 1954) is, however, hardly comparable to the Batagay YIC. In contrast to Batagay, the Mus-Khaya Ice Complex is affected by fluvial deposition resulting in a cyclic facial-lithological structure represented by dark-brown, organic-rich,



loess-like loam alternating with dark-grey, ice-rich loam. This alternation of organic-rich and ice-rich sediments of different composition is the basis of the cyclic YIC structure theory, because the deposits are believed to be of predominantly alluvial origin (Katasonov, 1954; Lavrushin, 1963; Popov, 1967). This theory can be well implemented for floodplain settings, because the cycles represent changes, governed by shifts in the river course, from riverbed to oxbow lake and floodplain

deposits. Such cyclic structure is not detectable at the Batagay outcrop because this site was not affected by river influence as it is distant from a river floodplain. On the contrary, the absence of such cyclic structure indicates the slope genesis of the studied YIC.

Mamontova Gora is situated along the Aldan River in central Yakutia, outcropping in a 50-m high terrace (Markov, 1973). It was stratigraphically subdivided into 3 units covering the time span from Holocene to presumably Last Interglacial (Pewe et.

al., 1977). The middle unit of the Aldan River outcrop revealed radiocarbon ages from 26 ka BP to >56 ka BP, which correlates to unit II in Batagay sequence and invites the assumption that an erosional event took place in central Yakutia on a similar temporal scale as in the Yana Highlands. The main difference in the Mamontova Gora sequence to the Batagay outcrop is the dominance (60%) of well sorted silt with grain size values of 0.005-0.5 mm, which was explained by distant aeolian particle transport from wide, braided, unvegetated flood plains of rivers draining nearby glaciers (Pewe et. al., 1977).

Most of the coastal permafrost exposures in Siberia are characterized by bimodal or polymodal GSD curves (Schirrmeister et al., 2008; Schirrmeister et al., 2011b), which indicates a mixture of transport, accumulation and re-sedimentation processes occurring there. Unimodal and bimodal curves, as were revealed for both, Mamontova Gora and the Batagay mega thaw slump, could reflect more stable accumulation and sedimentation processes under continental conditions.

The Batagay outcrop is one of the few permafrost profiles accessible in interior Yakutia. The present study offers rare insights into the evolution of northern environments under the conditions of the most severe climatic continentality in the Northern Hemisphere. Further studies should be focused on cryolithological material from units IV and V to fill the current gaps in knowledge about the formation of the Batagay sequence. Sedimentological and cryolithological analyses and the study of fossil bioindicators (plant macro-fossils, pollen, insects, and mammal bones) will contribute to the reconstruction of

Quaternary palaeoenvironments in Western Beringia.

**5 Conclusions**

•       The Batagay mega thaw slump is one of the few active permafrost outcrops in interior Yakutia, which provides rare insights into sedimentation processes, climate and environmental evolution under the conditions of the most severe climatic

continentality in the Northern Hemisphere.

•       As indicated by OSL-dates >200 ka (infinite), the exposed sequence was deposited over a large time span at least since the Middle Pleistocene.





- Altogether five distinct sedimentological units, representing different accumulation phases, were detected (top-down): a Holocene cover layer, the late Pleistocene Yedoma Ice Complex, an organic horizon deposited during the last interglacial, a thick, banded, uniform unit without visible ice wedges, and another Ice Complex older than the last interglacial.

• The detected five cryolithological units reveal distinct phases in the climate history of Interior Yakutia: the existence of a middle Pleistocene Ice Complex indicates cold stage climate conditions at the time of deposition of Unit V resulting in a mean annual ground temperature at least 8 °C lower than today (Romanovskii et al., 2000b).

- A climate shift during deposition of Unit IV caused cessation of ice wedge growth due to highly increased sedimentation rates and eventually caused by a rise of temperature.

• Full interglacial climate conditions existed during accumulation of the organic-rich Unit III. In contrast to other MIS 5e deposits in Yakutia, e.g. in the coastal lowlands, no plant or mollusc remains indicating aquatic or palustrine environments could be detected. On the contrary, plant macrofossils reflected open forest vegetation existing under dry conditions during the last interglacial.

- The late Pleistocene YIC (MIS 4-2) occurring in Unit II proves again severe cold-stage climate conditions with a
mean annual ground temperature 8 to 10 °C lower than today.

- Peatland deposits as indicators for thermokarst processes as they are characteristic for Early Holocene sites in the Circumarctic (MacDonald et al., 2006) were not detected at the Batagay outcrop. This might be due to the absence of intense thermal degradation or due to the topographical setting preventing melt water to accumulate.

- As is indicated by radiocarbon AMS dating, gaps in the sedimentological record are existing likely as a result of
erosional events or of spatially and temporarily differential small-scale deposition.

- Compared to other YIC sites in Yakutia, Unit II of the Batagay profile could be classified as a 'highland' type of YIC, which is characterized by its geographical position distant from rivers and sea coasts and in proximity to hills and mountains more inland. Whereas fluvial and lacustrine influence is common for certain depositional periods in the majority of permafrost exposures on the Yakutian coastal lowlands, it has to be excluded for the Batagay sequence.

• We suggest that the prevailing sedimentation processes and the sources for the deposited material varied seasonally during the formation of the YIC in the Batagay profile.

**Author contribution**

F.K. designed the study conception, arranged the expedition. F.K. and K.A. carried out field description and sampling. L.S.
accomplished the sedimentological analysis and plotted the graphs. M.F. designed and performed the OSL dating procedure and interpretation. K.A., L.S., F.K., M.F., participated in drafting the article. K.A. prepared the manuscript with contributions from all co-authors. F.K., L.S., K.A. revised the draft. Authors give final approval of the version to be submitted.



**Competing interests**

The authors declare that they have no conflict of interest.

**Acknowledgments**

This research was funded by the German Science Foundation (DFG, KI 849/4-1) and supported by the German Federal Ministry of Education and Research (BMBF, Arc-EcoNet, 01DJ14003). We feel sincere thanks to Dr. Elena Troeva from the Institute for Biological Problems of the Cryolithozone Yakutsk, who provided great support in coordination, planning and in logistical processes. We thank Russian colleagues for great support during fieldwork in Batagay, especially for the patience and never-ending help in transport by Vladimir Malyshenko, and for the hospitality of Anna Jumshanova and Sargylana

Sedalischeva and their willingness to organize a small home lab. Thanks to Prof. Dr. Ludmila Pestryakova from the Northeast Federal University Yakutsk and her colleagues for help with the sample logistics. We gratefully acknowledge prompt work on dating under the supervision of Tomasz Goslar at the Poznan Radiocarbon Laboratory, Poland. We are grateful for the helpful and constructive comments on the manuscript by Dr. Christine Siegert and Dr. Sebastian Wetterich from the Alfred Wegener Institute, Helmholtz Centre for Polar and Marine Research Potsdam. The analytical work in AWI

laboratories was expertly conducted by Dyke Scheidemann.

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



Table 1. Cryolithological description of the Batagay permafrost sequence.

| Unit | Section | Observed depth (m bgs) | Field description |
|---|---|---|---|
| **I** | A | 0-0.09 | Sod, no ice, composed mainly of modern plant litter including living plant parts. |
| | | 0.09-0.2 | Light brown sediment with dusty structure. No ice. Horizon is penetrated by modern roots. |
| | | 0.20-0.43 | Homogeneous light brown layer. No ice. Inclusions of oxidized iron and charcoal. Black spots 30-45 cm deep indicate relocation of solutes and incipient new mineral formation. The border to the underlying sediments is straight and horizontal. |
| | | 0.43-0.85 | Brown horizon. No ice. Enriched with charcoal and modern plant roots. |
| | C | 0.0-1.4 | Silty sediments of dark grey colour, inclusions of charcoal. |
| | | 1.40 | Top of an ice wedge. |
| **II** | A | 0.85-4.0 | Sandy-silt in layered ice, layers of gravel, few plant remains, roots. |
| | | 4.60-4.72 | Reddish-colored horizon with 8-cm-wide ice veins crossing vertically. Rich in plant remains, contains an arctic ground squirrel burrow. |
| | | 5.0-5.8 | Unstructured grayish sandy silt with abundant plant remains. |
| | | 5.8-6.5 | Dark grey ice-rich sandy-silt. |
| | | 6.5-9.5 | Horizontal layers of greyish-brown sand (up to 7 cm thick) and ice bands (up to 5 cm thick); borders are well pronounced, sharp. No visible plant material. |
| | C | 10.0 | Sandy silt, horizontal layered ice bands. No visible plant material. |
| | | 16.5 | Brownish-grey sandy silt, less ice-rich than above. Inclusions of plant roots. |
| | | 19.5 | Light brown horizon dissected by horizontal to sub-horizontal ice layers. Alternation of clayey and sandy layers with distinct wavy borders. |
| | | 22.0 | Fulvous brown horizon with 1-mm-thick ice veins. |
| | | 24.5 | Homogeneous strata of grayish sediment structure, less ice. Distinct color border with the underlying horizon. |
| | | 32.0-32.5 | Brownish-yellow horizon with abundant plant remains. |
| | | 32.5-37.0 | Homogeneous strata of grayish sediment, horizontally-layered ice bands. |





| | | 37.0-37.5 | Alternation of grey and black layers, the latter with fulvous inclusions. |
|---|---|---|---|
| | | 37.5-43.5 | Layered brown sediments. |
| III | B | 40-42 | Alternation of sandy-silty layers with plant remains. Frozen organic sediments are extremely rich in large macroscopic plant remains including numerous branches and twigs of woody plants. |
| IV | B | 42.0-50.0 | Layered brown sands, no ice wedges. |
| V | Bottom in the central part of the thaw slump | | Thick vertical ice wedges and dark layered sediment columns. |

Table 2. Radiocarbon dating of selected samples from the Batagay permafrost exposure.

| Lab. No. | Sample name | Depth [m bgs] | Section/ Unit | Radiocarbon ages [ka BP] | Calibrated ages 2 σ 95.4% [cal ka BP] | Description |
|---|---|---|---|---|---|---|
| Poz-78149 | 19.6/A/4/1.15 | 1.15 | A/I | 0.295±0.03 | 0.459 –0.347 | Plant remains |
| Poz-79751 | 19.6/A/5/2.05 | 2.05 | A/IIc | 33.400 ± 0.5 | 37.305 – 38.259 | Plant remains |
| Poz-77152 | 20.6/A/1/460-472 | 4.6 | A/IIc | 26.180± 0.22 | 28.965 – 27.878 | *Plantago* sp., *Artemisia* sp., ground squirrel droppings |
| Poz-79756 | 22.6/C/2/8.5 | 8.5 | C/IIc | 12.660 ± 0.05 | 14.919 – 15.209 | Plant remains |
| Poz-79753 | 22.6/C/6/12.5 | 12.5 | C/IIc | >48.00 | | Plant remains |
| Poz-79754 | 22.6/C/9/14.5 | 14.5 | C/IIc | >51.00 | | Plant remains |
| Poz-79755 | 29.6/E/2/18.5 | 18.5 | C/IIb | 46.00 ± 2 | 49.034 – 50 | *Papaver* sp. |
| Poz-78150 | 29.6/C/1/24.5 | 24.5 | C/IIb | 110.31± 0.37pMC | 1991AD - 2005AD | *Alnus* sp., *Vaccinium vitis-idea* |
| Poz-66024 | 21.6/B/3/2 | 44 | C/III | >44.00 | | Plant remains |



Table 3. OSL and IRSL measurement data and respective dating results for the luminescence samples from unit IV of the Batagay permafrost exposure (Dose rate: effective dose rate calculated based on results from gamma-spectrometry, cosmic dose rate and corrected for mineral density, sediment density, grain sizes and water content; Water: in-situ water content/saturation water content; N: number of aliquots; PD: paleo-dose based on central age model, CAM, according to Galbraith et al., 1999; OD: overdispersion, Age: calculated ages according to CAM using the in situ water content, > indicates minimum age signals were close to saturation and hence, tend to underestimate luminescence ages)

| Sampling site | N 67° 39′ 18″, E 134° 38′ 30″, 280 m asl | | | | | | |
|---|---|---|---|---|---|---|---|
| Sample name | Depth [m] | Water [%] | Dose rate [Gy/ka] | Grain size [μm] | N | PD (CAM) [Gy] | OD [%] | Age [ka] |
| *QUARTZ* | | | | | | | |
| 2.7/B/1/47 | 47 | 30.1/49.6 | 1.3 <br> 1.4 | 90-160 <br> 63-100 | 26 <br> 19 | 123.8 ± 6.2 <br> 129.0 ± 6.1 | 26.5 <br> 17.1 | > 93.6 <br> > 95.2 |
| 2.7/B/2/47 | 47 | 34.3/51.6 | 1.3 <br> 1.3 | 90-160 <br> 63-100 | 11 <br> 11 | 127.1 ± 5.1 <br> 185.3 ± 26.1 | 6.6 <br> 42.9 | > 100.2 <br> 142.8 ± 25.3* |
| 2.7/A/2/50 | 50 | 25.1/37.4 | 1.4 | 63-100 | 12 | 174.4 ± 14.4 | 23.7 | > 123.2 |
| *FELDSPAR* | | | | | | | |
| 2.7/B/2/47 | 47 | 34.3/51.6 | | 63-100 | 25 | 274.2 ± 3.32 | 3.9 | 210.0 ± 23.0** |

\* The CAM age using the saturation water content yields 160.9 ± 27.7 ka

\*\* The  age using the saturation water content yields 236.6 ± 24.0 ka



Table 4. Overview of permafrost dynamics recorded in the Batagay sequence in correlation with global and regional climate histories. Global climate history is represented by Marine Isotope Stages (MIS; Aitken & Stokes, 1997) derived from the δ¹⁸O curve (modified from Pisias et al., 1984), reflecting global temperature changes studied in deep sea cores: negative δ¹⁸O ‰ reflect warm climate stages, while positive values identify cold phases. The regional Siberian climate phases are given according to Sachs (1953); The European regional climate events for comparison are named according to Litt et al. (2007).

| δ¹⁸O (‰) | Date BP, ka | MIS | Siberian classification | European classification | Unit | Permafrost dynamics |
|---|---|---|---|---|---|---|
| | < 11.5 | 1 | Holocene | Holocene | I | Permafrost degradation, erosional processes |
| | 28 – 11.5 | 2 | Sartan stadial | Late Weichselian | IIc | Yedoma Ice Complex, thickest ice wedges - coldest climatic conditions |
| | 50 - 28 | 3 | Kargin interstadial | Middle Weichselian | IIb | Yedoma Ice Complex with warm phase signals –TOC values higher than in units IIc and IIa |
| | 73 - 54 | 4 | Zyryan stadial | Early Weichselian | IIa | Yedoma Ice Complex agradation – thick ice wedges, low organic content |
| | 120 - 127 | 5 | Kazantsevo interglacial | Eemian | III | Thick organic layer, warmest period within the sequence |
| | > 130 | 6 | Taz stadial | Late Saalian | IV | Cessation of Ice Complex formation, increased sedimentation rates, shift in climatic conditions |
| | | | | ? | V | Middle Pleistocene Ice Complex, thick ice wedges, cold stage climate |



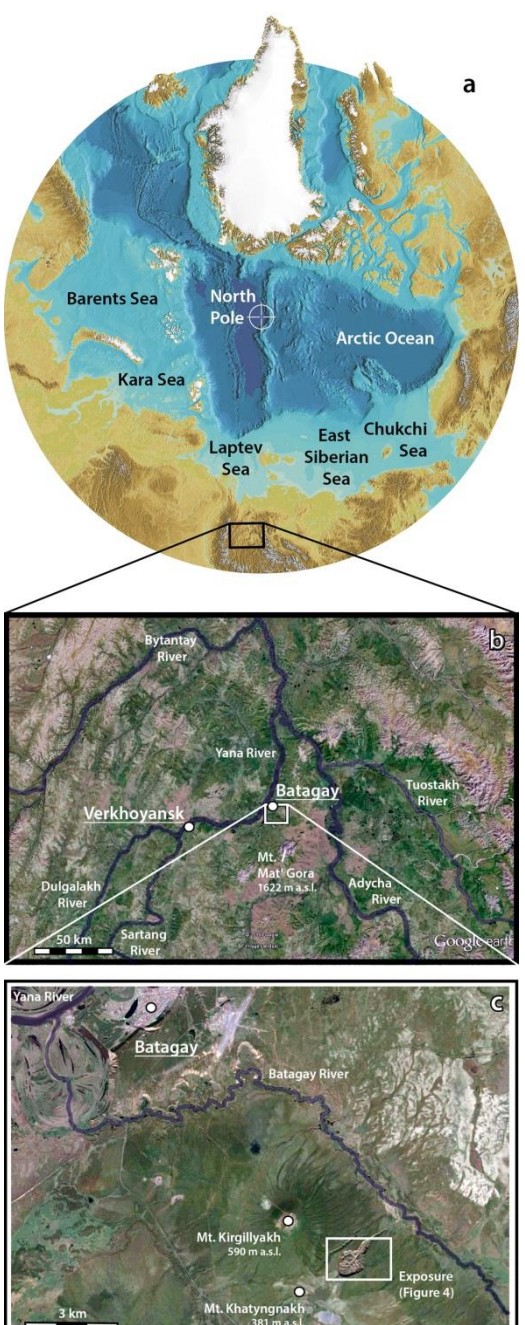

**Figure 1: (a) Location of the Yana Highlands in northeastern Siberia. Map modified from the International Bathymetric Chart of the Arctic Ocean (Jakobsson et al., 2012). (b) Situation of the study area on the right southeastern bank of the Yana River valley. (c) Location of the Batagay mega slump (framed) at the northeastern slope of Mt. Khatyngnakh, left bank of the Batagay River. (b) and (c) modified from satellite pictures, Google Earth V. 7.1.2.2041. (July 4th, 2013), Batagay Region, Russia, 67°34'41.83''N, 134°45'46.91''E, Digital Globe 2016, CNES Astrium 2016, http://www.earth.google.com (accessed April 25th 2016).**



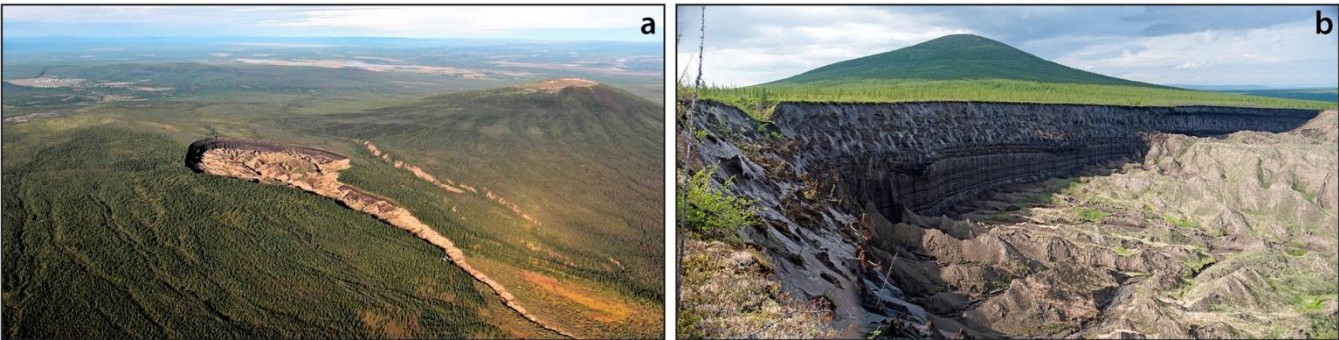

**Figure 2: General views of the Batagay mega slump. (a) From aircraft (L. Vdovina, Yana Geological Service, August 17th, 2011). (b) The exposure at its deepest incision photographed from the southern edge of the cirque (June 19th, 2014). For orientation, note Mount Kirgillyakh in the upper right (a) or in the background (b).**

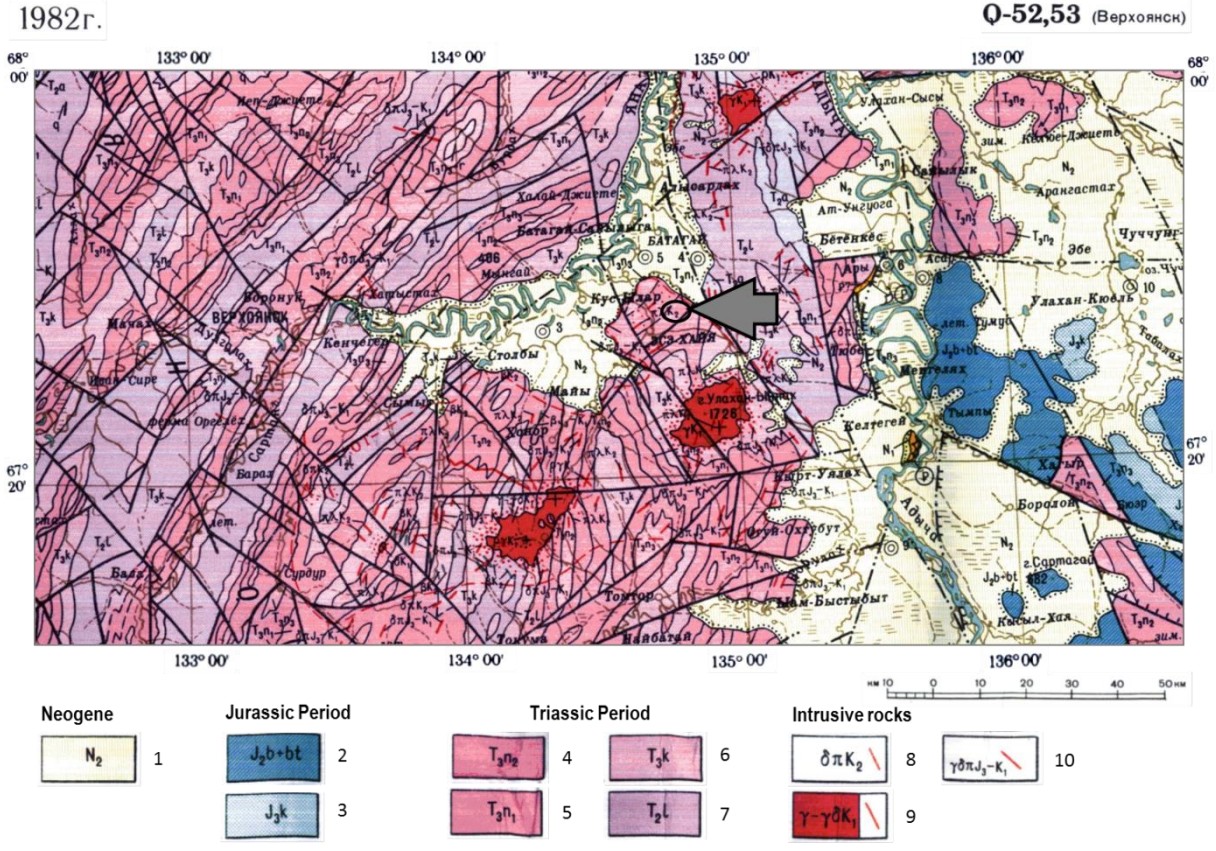

**Figure 3: Geological map of the study area. A black circle and a grey arrow indicate the Batagay thaw slump location. 1 - Pliocene. Gravel, sandstones, loams, siltstones, conglomerates, peat. 2 - Bathonian and Bajocian stages. Sandstones, siltstones, mudstones, calcareous sandstones. 3 - Callovian stage. Sandstones, siltstones, mudstones, calcareous sandstones. 4 - Middle Norian stage. Siltstones, mudstones, sandstones and conglomerates. 5 - Lower**
10 **Norian stage. Siltstones, sandstones, mudstones, tuffaceous rocks, conglomerates and gritstones. 6 - Carnian stage. Mudstones, siltstones, sandstones, conglomerates. 7 - Ladinian stage. Sandstones, siltstones, conglomerates. 8 - Late Cretaceous period. Dikes of diorite porphyry ($\delta\pi K_2$) and quartz porphyry ($\pi\lambda K_2$). 9 - Early Cretaceous period.**





**Biotite and binary granites (ϒK₁). 10 - Late Jurassic- early Cretaceous period. Granodioritic-porphyry (ϒδπJ₃-K₁), dioritic-porphyry (ϒπJ₃-K₁), quartz-porphyry (πλJ₃-K₁). State Geological Map USSR, page Q-52,53, 1982, modified.**

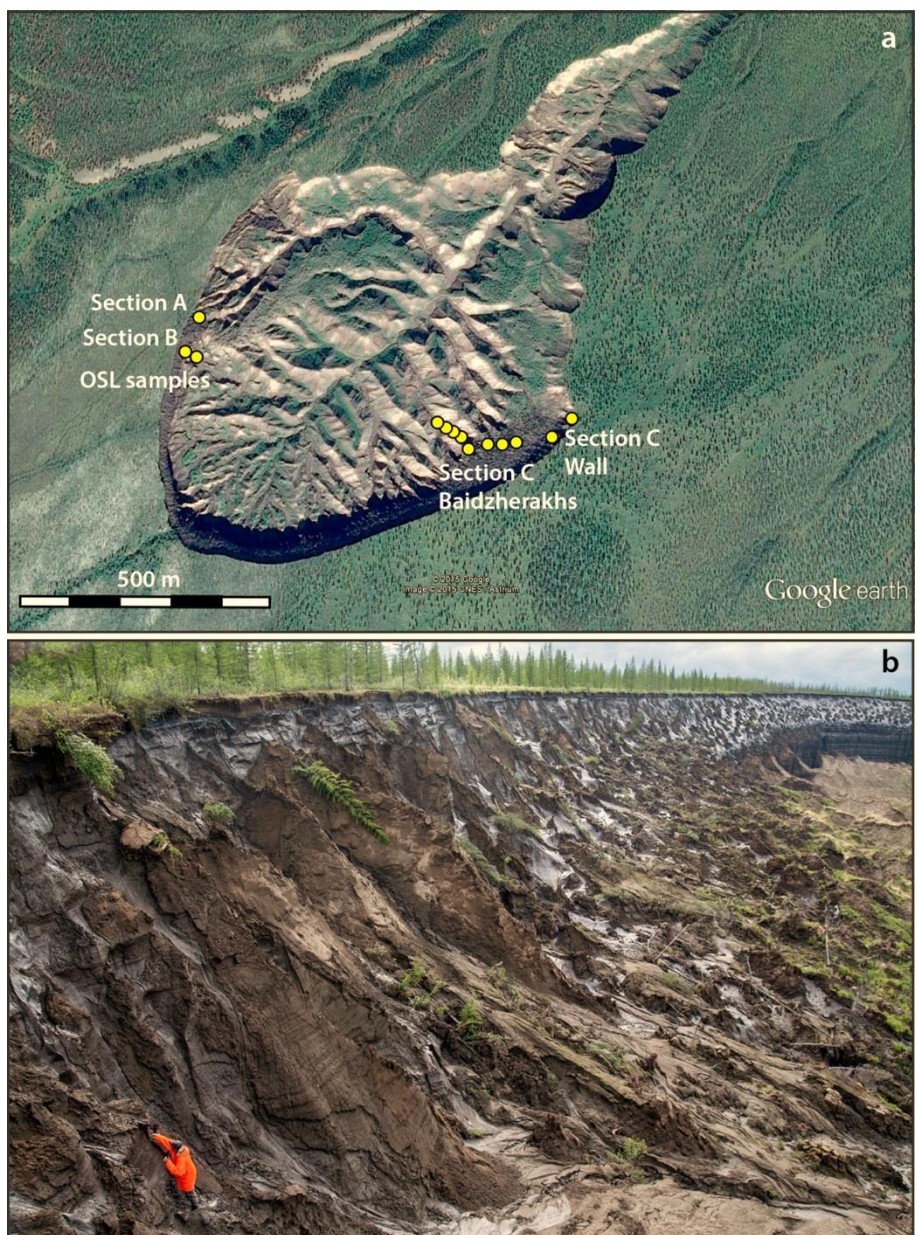

Figure 4: (a) Location of the studied sections in the Batagay mega slump. Modified from a satellite picture, Google Earth V. 7.1.2.2041. (July 4th, 2013), Batagay Region, Russia, 67°34'41.83''N, 134°45'46.91''E, Digital Globe 2016, CNES Astrium 2016, http://www.earth.google.com (accessed April 25th 2016). (b) Southeastern slope of the thaw slump; section C during sampling. Note person for scale.



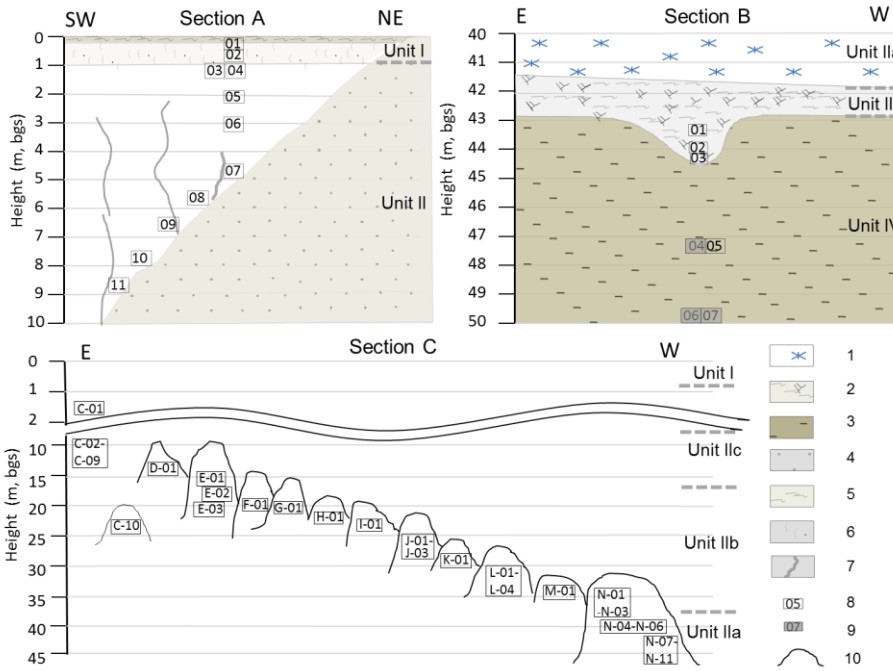

**Figure 5: Sections of the Batagay permafrost exposure. 1 – ice-rich sediments; 2 – organic layer with plant remains; 3 – layered cryostructure; 4 – sand; 5 – plant detritus; 6 – active layer with roots and coal; 7- ice wedge; 8 - sediment and macrofossil sample site; 9 – OSL and sediment sample site; 10 – baidzherakh.**



**Figure 6: The cryolithological structure of the Batagay exposure in its western and southwestern part. (a) General position of the detected cryolithological units (I to V). (b) Overall view of the outcrop. (c) Unit I (140 cm thick active layer) and boundary to Unit**





II (YIC) in Section A. (d) Unit II, steep wall of the YIC illustrating the three observed subunits differing in ice content and contour. The trees as scale on top of the wall are about 6-8 m tall. Section A is situated at the upper part of the slope, on the right side of the photo. (e) Detail of the three lower cryolithological units III, IV and V. The old Ice Complex Unit V with preserved syngenetic ice wedges is only partly exposed.





**Figure 7: Typical sediment and cryostructures at the Batagay exposure. (a) Contact zone between the active layer, Unit I, and YIC, Unit II (section C). (b) Charcoal inclusions and iron oxide impregnations in Unit I (section A) at 0.20-0.43 m bgs. (c) Horizontally-layered cryostructure of Unit II (section C). (d) Fossil ground squirrel nest (dated ca 26 ka BP) at 4.7 m bgs in Unit II (section A). (e) Organic-rich deposits filling an ice wedge cast ca. 42 m bgs in Section B; the person illustrates the position where sample Nr. 21.6/B/1/43 was taken. (f) Sample Nr. 21.6/B/1/43 in frozen state showing alternate bedding of sand and plant detritus layers. Thickness of the upper plant detritus layer is about 5 cm. (g) Ice-rich deposits in layered cryostructure enclosed by several m thick syngenetic ice wedges in Unit V.**

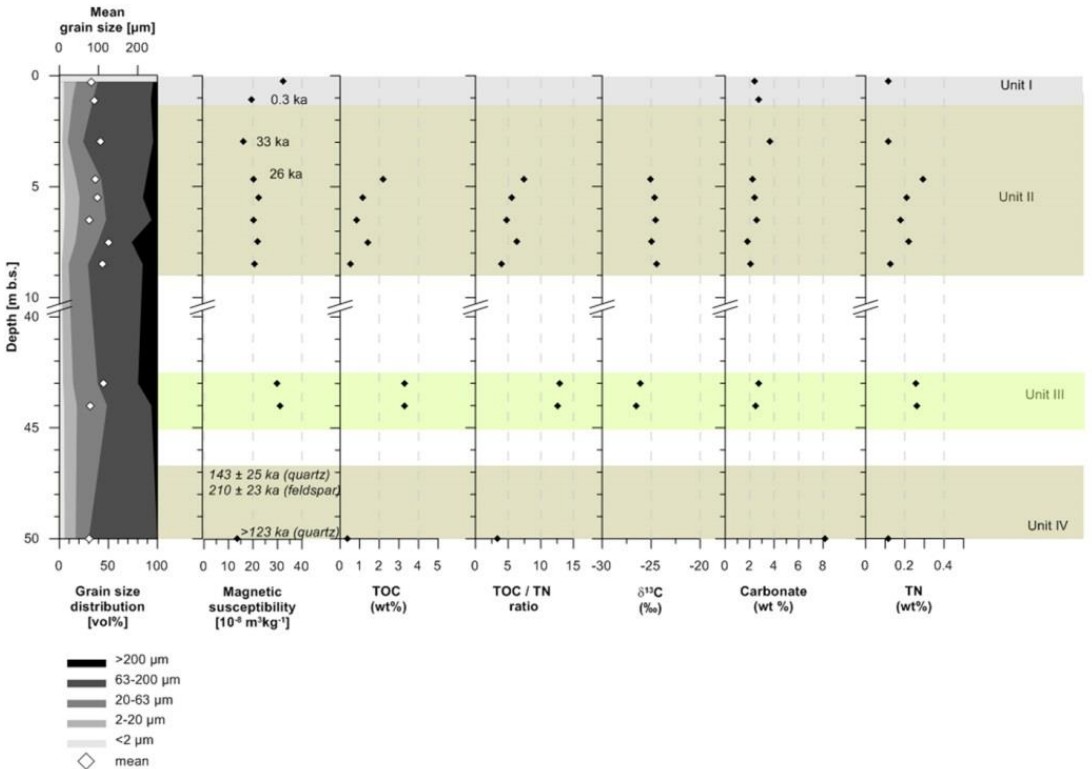

**Figure 8: Diagram presenting grain size distribution, MS, TOC and TOC/TN, δ¹³C and carbonate records for section A.**





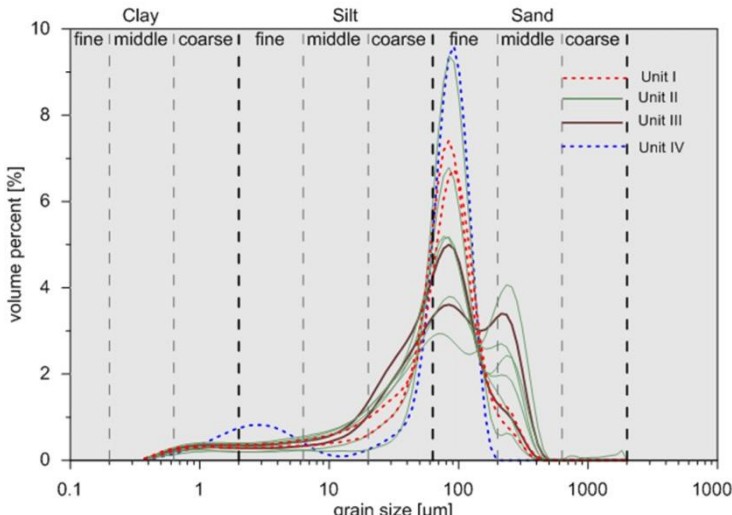

**Figure 9: Grain size distribution plot for sections A and B of the Batagay permafrost outcrop.**

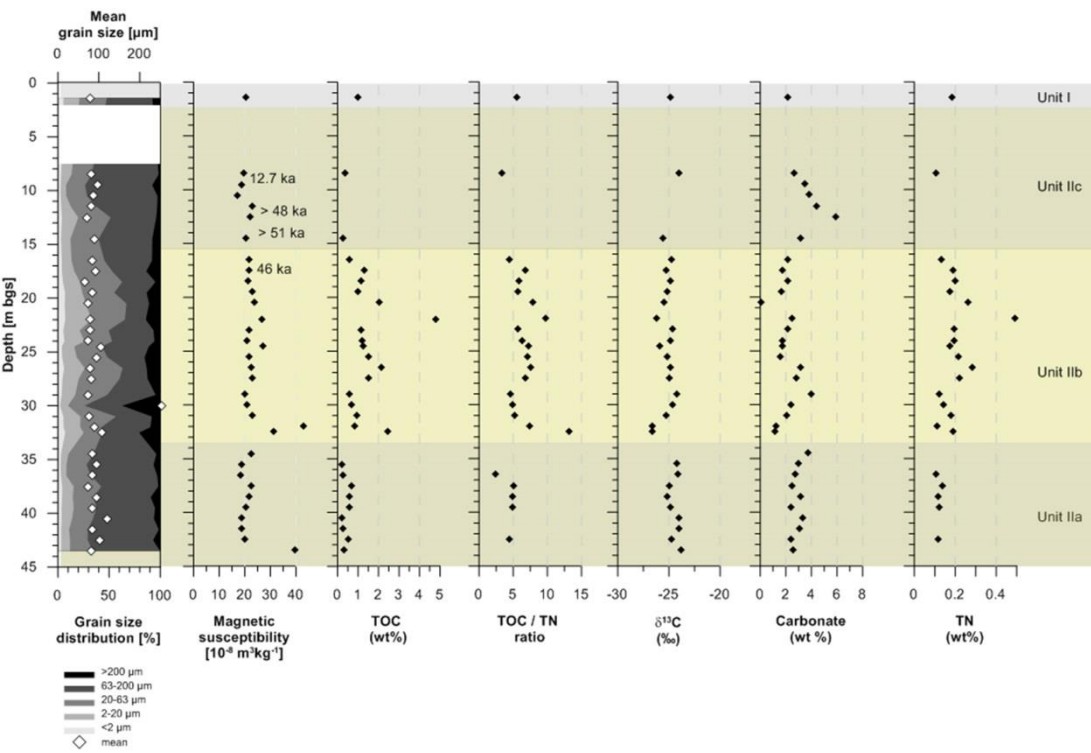

5   **Figure 10: Diagram presenting grain size distribution, MS, radiocarbon ages, TOC, TOC/TN, δ¹³C and carbonate records for section C.**



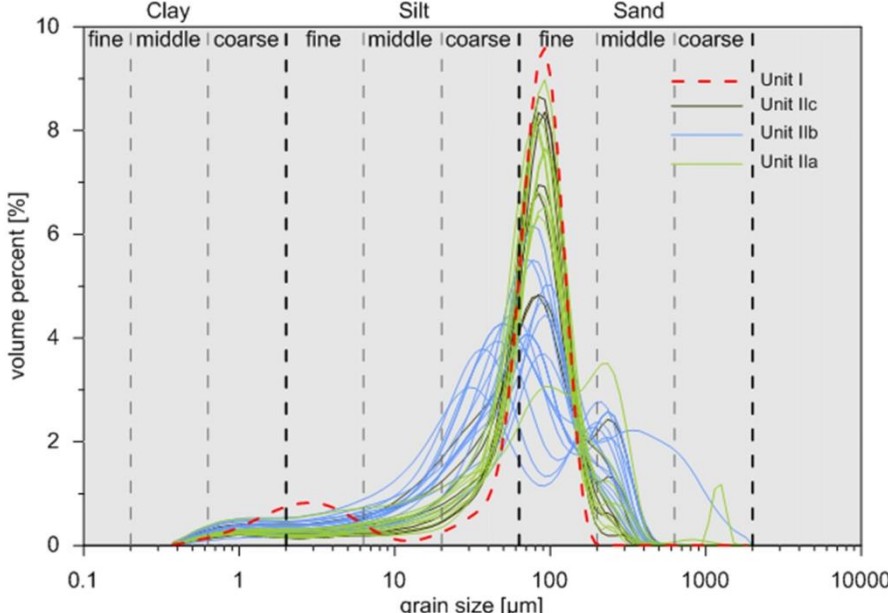

Figure 11: Grain size distribution plot for section C of the Batagay exposure.