# Peer review of "Paleoclimate characteristics in interior Siberia of MIS 6-2: first insights from the Batagay permafrost mega thaw slump in the Yana Highlands"

_Climate of the Past, 2016_

## Short Comment (SC1) · 18 Dec 2016

**Review of: Ashastina et al., Batagay Megaslump Exposure, Climates of the Past: Discussions**

**General Comments**

This paper describes the stratigraphic sequence exposed by the Batagay mega slump headwall in the Yana Highlands region of Siberia. Several sections representing different positions in the exposure were described and sampled for paleobotanical, cryostratigraphic, and geochemical analysis. The dating is based on nine radiocarbon ages, and three OSL dates. The section represents at least 142 ka of discontinuous deposition based on the lowest OSL age and several erosional unconformities. The authors discuss the geomorphic and paleoclimatic implications of this record, and how this section relates to other permafrost exposures in Siberia.

In general, I question whether this paper is appropriate for publication in *Climates of the Past: Discussions* because it's main focus seems to be the detailed interpretation of the depositional environment at the Batagay sequence. The climatic connections are not strongly made until well into the discussion, and need to be developed much further if this record is to be published. For example there is very little introduction to the paleoclimate issue being addressed in this study.

In addition, I found there to be a lack of description of the sedimentary facies in the section, which could greatly aid in the interpretation and back up some of the claims the authors make. I suggest that the authors outline the methods and references used for identifying different depositional environments. These should be based on modern-day analogues with pictures of the modern-day depositional processes and their facies compared with these corresponding facies in section. In general the final section of the Discussion makes several assertions about the origin of the sediment without such comparisons.

I also find the uncertainties behind the low-resolution dating, 14C age reversals, and poorly constrained erosional unconformities to prevent the firm connection of this record to global climate proxies and regional sequences that are dated at higher resolution and have continuous deposition.

One of the main themes of this paper is to address the ongoing controversy of what depositional process generated the yedoma ice complex. I think this is a very interesting and relevant issue, but I don't think this journal is the appropriate venue for this. If the editor feels that this is not the case, then I highly suggest that the interpretation of the depositional environments at Batagay be developed much further with more evidence from modern environments and a better description of the sedimentary facies in the exposure. I elaborate on many of these points in my detailed comments below.

**Detailed Comments**

Page 1 Line 14: Late Pleistocene should be capitalized.

Page 1 Line 18: '…sought climate record.'  Should be reworded.

Page 1 Line 19: 'close by the pole of cold' I am not sure this name is well known.

Page 1 Line 25: '8° C colder than today' What is this quantified reconstruction of MIS 8 based on? Are they talking mean annual temperature? See below for more on this.

Page 1 Line 28: 'proves again' should be reworded.

Page 1 Line 30: 'In the Holocene cover….' I think the authors mean in the Holocene unit.

Page 2 Lines 6-10: This final statement of the abstract is one side of an ongoing controversy about the origin or processes that generate yedoma deposits. If the authors are going to interpret such deposits as being formed by nival and proluvial processes, then I think they should briefly describe the basis for this interpretation.

Page 2 Lines 22-25: This portion of the introduction describes the controversy of what geomorphic process is the cause of the yedoma ice complex. The authors are questioning whether aeolian deposition was primarily responsible because 'there existed a diversity of habitats, including aquatic'. I do not understand why the existence of aquatic habitats precludes the aeolian interpretation. Permafrost can perch the water table near the surface and this can create aquatic habitats in otherwise dry environments. I think the authors may want to describe better the basis for why they are questioning the aeolian hypothesis, and how this study can address this controversy. In addition, this controversy seems to be the main theme of the paper, and not in line with the Climates of the Past Discussions Journal

Page 3 Line 32: 'Globally greatest temperature gradient' should be reworded.

Page 4 Line 4: 'Accepted as the lowest temperature in the Northern Hemisphere' If there is a citation for this, then it should be called here.

Page 4 Line 11: 'Resembling' should be 'Similar'

Page 5 Line 12: 'Possible reservoir effects as a result of the accidental use of freshwater aquatics…' This does not make sense. How did the authors know the macrofossils were aquatic? Did they identify them as such or did they infer this based on the $\delta^{13}C$ values. In addition, the authors should indicate what $^{14}C$ calibration curve was used.

Page 5 Line 16 to Page 6 Line 7: I am not an expert in OSL dating, but the methods described here seem to follow standard techniques in the literature.

 Page 6 Line 10-16: It seems unnecessary to describe, in detail, how the thaw slump is positioned and behaves to start off the results section. This section does not seem to have much bearing on the main points of the paper. If this section needs to be included in the

paper, then I suggest it go in the Study Site section. If the point of this is to say that the depths of different sections of the slump cannot be compared because some of them are not vertical, then this could be reduced to a few sentences.

Page 6 Line 22: I assume these meter calls are being measured from the top of the slump. This should be specified here.

Page 9 Line 26: The authors think that the sedimentary transitions of the different units represent erosional unconformities. Do they see cut and fill or other features to back this up? I do not doubt this interpretation, but it would be useful to describe the reasoning behind this. In my opinion this interpretation can be in the Results.

Page 9 Line 31: I find the 300-year BP $^{14}$C age on plant material that is 1.15 m below the surface to be suspect. Is there loess deposition in this region today? How could 1.15 m of sediment accumulate in 300 years without incredible rates of productivity, a mass movement above the section, or high rates of loess deposition? Bluff-top sequences of loess in section often have reworked loess that blew onto the ground surface as the cut-bank neared the site of the section. Is this a possibility? It would seem more likely that this date represents modern material from rooting or cryoturbation from the current vegetation mat.

Page 10 Lines 1-8:

Unit II corresponds to the YIC. YIC deposits can form only under extremely cold winter conditions. They are thus indicative of cold stage climate in a continental setting. Our dating results confirm the assumption that the YIC was deposited from at least >51 ka BP to 12 ka BP, thus during the last cold stage and including the MIS 3 (Kargin) interstadial period. Huge syngenetic ice wedges and high segregation ice contents are the most typical features of Yedoma sequences. The structure of ice wedges intersecting sediment columns are evidence for the syngenetic freezing of the ice-wedge polygon deposits. The ice wedges were 4.5 to 6.5 m wide, which indicates the impact of an extremely cold climate during their formation and also indicates aridity (Kudryavtseva, 1978). The thermokarst mounds (baidzherakhs) appearing in staggered order 4.5-6.5 m apart on the upper southeastern part of the YIC support this hypothesis.

The dating results from this study do not necessarily indicate that the YIC was deposited continuously from 51-12 ka. This is because there are only a few 14C ages from this unit and they seem to be subject to reworking. Could an alternative view be that YIC accumulated only episodically or during a fraction of this time period because the plant remains were reworked or the deposits are too coarsely dated to infer continuous deposition.

Page 10 Lines 25-27: The authors say that the 14C age reversal could be due to a ground squirrel stashing food underground, which would bring younger C down below older C in section. The two dates are 26.2 and 33 ka. The younger date is 2.55 meters below the older one. Because ground squirrel food caches are limited by permafrost (they have not been observed to burrow into frozen ground), this would suggest that the active layer at this site was at least 2.55 meters deep. This does not seem plausible. The authors should discuss this further if they think it to be possible.

Page 11 Lines 1-5: The authors describe how the erosional unconformity probably corresponds to a thermal erosion event during the warm times of the Pleistocene Holocene transition. This may be true, but it should be acknowledged that the $^{14}$C dates that bracket this erosion event seem to span around ~25.7 ka. The Bolling Allerod and early Holocene warm interval were millennial-scale events. I think it should be acknowledged that this correspondence is highly speculative given the age control.

Page 11 Line 22: The authors say that the climatic conditions were insufficient for ice wedge growth, but climate is only part of driver for ice wedge growth. The type of depositional environment and grain size of Unit IV could also prevent ice wedges from forming or being preserved. The authors should rule out whether non-climatic factors contributed to the lack of ice wedges in this unit.

Page 12 Line 1: The authors state that the presence of ice wedges in Unit V indicates that the mid-Pleistocene was characterized by extremely cold winters. This statement does not seem to be based on any dating, and relies on stratigraphic order. It should be acknowledged that just because the ice wedges are below the MIS 5 paleosol that this Unit V does not necessarily represent the Mid-Pleistocene. Similarly, the authors state that the ground ice in Unit V survived multiple interglacial warm times, but they only show that the ice survived MIS 5.

Page 12 Line 25-30: The authors state that the only mechanisms for the deposition of >50% sand in Unit II come from proluvial, nival, or periglacial processes, but give no citation. The authors do not think that aeolian processes could be responsible for depositing this unit. In many aeolian settings, silt and sand can be deposited together depending on sediment availability, wind energy, and surface roughness. It is not uncommon to have sand sheet interbedded with loess deposition. A more detailed report on the sedimentary facies in the section could constrain whether aeolian processes are at play here.

Page 13 Lines 13-15: A citation call would be useful to back up the interpretation that changes in magnetic susceptibility is a prerequisite for aeolian deposition here. I think there could be little change in MS under varying aeolian processes.

Page 13 Lines 17-20:

genesis, as was discussed for subunits IIa and IIc. Nevertheless, the high percentage of the silt fraction in the GSD of subunit IIb cannot be interpreted as an exclusive indicator of aeolian deposition, because high silt content in the sediment composition can also result from cryogenic disintegration of quartz due to repeated thawing and freezing cycles (Konishchev and Rogov, 1993; Schwamborn et al., 2012).

It is not clear to me why free-thaw action on quartz grains excludes the possibility of aeolian deposition here. Wouldn't freeze-thaw action be prevalent in this region regardless of the climate or depositional environment?

Page 13 Lines 29-30: The authors call MIS 5a the last glacial period. This seems too similar to the common name for MIS 2, which is often called the last glacial period. I suggest another name.

Page 14 Line 12: The mean annual ground temperature is only partly driven by climate. Surface processes, like the thermal conductivity of different soils and the thickness of the insulating snow layer, should be discussed as these features were likely different in the past.

**Abstract in General:**

In general, the abstract is too long. It should be cut in half to describe the main motivation, approach, and points of the study.

The order of the abstract is counterintuitive to the study. First, the authors introduce the site, and units with some specific temperature reconstructions. Then the authors describe detailed methods that they used including the sampling interval. These methods shouldn't be in the abstract, and certainly should not come after the main points of the paper are described.

Similarly, details about organic carbon magnetic susceptibility etc. do not need to be in the abstract if they are not contributing anything about the main points of the paper.

I ask that the authors reconsider describing the Siberian lowlands as a maritime climate. Potential evapotranspiration exceeds precipitation in much of the Arctic. Perhaps the authors mean that the region is less continental today than it was during glacial intervals when this yedoma deposit formed. The lowlands are also described as maritime in the Introduction.

**Introduction in General:**

The authors describe the climate of the Siberian lowlands as maritime and the study area in the Yana highlands as more continental. I suggest the authors include the mean climatic specifications to show how different the two regions are.

I also question whether these two sites *were* climatically different when they formed during past glacial periods. Because eustatic sea level was much lower, and permanant sea ice more extensive the whole region would have been more continental, and the lowlands would have been included in this. Therefore, the authors should describe how much different these areas were in the past.

The final few paragraphs in the introduction are better suited for the Study Site section as they describe the study site.

**Study Site in General:**

I suggest that the authors briefly describe the modern-day vegetation, and major geomorphic processes occurring in the region today aside from the slump.

**Results in General:**

In my opinion, it is not necessary to describe the angles of the bluff and sections at various depths. These are subject to change within a few days of being described and do not add much to the interpretation.

The results would read much better if this section were to be broken up into different units instead of different techniques. The authors could easily describe the lithology, chronology, organic geochemistry, paleobotany, etc of Unit I and then proceed to Unit II. This provides a narrative for what these units are composed of and when they were formed. I think this approach would also save significant space.

There is a distinct lack of information about sedimentary facies in this paper. The interpretation could be greatly aided by these results. Was there bedding or was each unit massive? What general attributes did these beds/laminations have? Was there fine rootlets embed in the sequence to suggest that the landscape was covered by vegetation? Was there any soil development? If so, what horizons / weathering is present?

The type of material that was 14C dated should be described in the text. 'Plant remains' should be specified to taxa.

It should also be specified how many aquatic plants were dated from this section, but not reported in this study. The methods give the impression that some 14C dates were excluded, but which ones, where were they sampled, and what were the ages?

**Discussion in General:**

The authors are assuming that the three sub-units in Unit II represent different marine isotope stages (MIS 4-2). Radiocarbon dating does not back up this assertion. It only seems to be based on the fact that there are three units and three isotope stages occurring at roughly this time. Much of the discussion on the possible links between the MI stages and subunits in Unit II could be removed.

The apparent erosional unconformities can be included in the Results if there are available sedimentary features that indicate where they are. As of now, most of the interpretation is based on large differences or reversals of 14C ages. This may not be warranted if the 14C dates are reworked, which the authors describe as a possibility.

The plant macrofossil identifications should be in the results.

I am skeptical that the current resolution of dating allows the statement that the Batagay sequence is in good agreement with global climate events over the last 125 ka. Mostly this section shows the landscape response to the last interglacial warm times, but there

are a number of other climate events since the mid-Pleistocene that may or may not be represented here. It is difficult to say with the unconformities and current dating resolution.

In my opinion, it makes much more sense to describe the depositional setting of the section prior to the paleoclimate interpretation.

The authors ignore the possible interpretation that much of the sediment was reworked from the nearby floodplain by aeolian processes and deposited into the uplands. Because many of these rivers have nival flow regimes, large areas of exposed sediment would have been available for aeolian transport. This has long been described as the mechanism to get loess into the uplands in many periglacial zones. I think it should be considered here as well.

The authors assert that proluvial and nival processes were at least partly responsible for the deposition of fine-grained material in the section. A description of the sedimentary facies could help back up this claim, but this is lacking. *I suggest that the authors outline the methods and references used for identifying different depositional environments. These should be based on modern-day analogues with pictures of the modern-day depositional processes and their facies compared with these corresponding facies in section.  In general the final section of the Discussion makes several assertions about the origin of the sediment without such comparisons.*

There are a number of other sections from around Siberia mentioned in the Discussion. It would be helpful to include the general locations of these places in the Map figures.

**Figures in General:**

The dating results should be much clearer then they are.

---

## Editor Comment (EC1) · D.-D. Rousseau (Editor) · 19 Dec 2016

Dear authors, Although apologizing for this late review, could you post a general reply to this very detailed review recently released about your manuscript? Thank you in advance. denis-didier Rousseau

---

## Author Comment (AC1) · 20 Dec 2016

Dear Dr. Benjamin Gaglioti,

Thank for your helpful and detailed feedback.

You are absolutely right with the resume of the manuscript: we focus on the description and interpretation of the depositional environments of the Batagay sequence, and discuss the palaeoclimatic implications of this archive.

The data discussed in our manuscript originate from a terrestrial permafrost archive and one of our goals is to check the link between the global /regional climatic changes during the Late Quaternary and response of the depositional processes to these changes. For this purpose, we compare the dated sections of the Batagay outcrop with sections of a similar age, described from other terrestrial permafrost outcrops in Northeastern Asia.

We find it appropriate to submit the manuscript to *Climate of the Past* journal because the manuscript suits one of the main subject areas of *Climate of the Past* ("reconstructions of past climate based on . . . data from marine and terrestrial (including ice) archives"). The Editorial board of *Biogeosciences* proposed that we submit the manuscript to *Climate of the Past*. In addition, the journal published a special issue on the El′gygygtgyn Impact Crater, with a set of articles focusing on sediment interpretation, depositional dynamics, and their climatic implications. A number of other articles published in this journal with the focus on sediment interpretation give us confidence to publish tour results in *Climate of the Past*.

We understand your concerns about the unconformities in dating results presented in the manuscript. Due to this reason we used several other proxies (description and comparison of the sedimentary facies, plant macrofossils) to connect dated parts of the record to dating results available from other regional permafrost sequences. We are confident that our stratigraphical interpretation is reasonable.

We appreciate your interest, detailed comments, and useful suggestions. We will take up your constructive comments to improve our manuscript.

---

## Referee Comment (RC2) · D. G. Froese (Referee) · 22 Dec 2016

Review of COPD "Pleistocene climate characteristics in the most continental part of the northern hemisphere: insights from cyrolithological features of the Batagay mega thaw slump in the Siberian Yana Highlands by K Ashastina, L Schirrmeister, M. Fuchs, and F. Kienast

Duane Froese Department of Earth and Atmospheric Sciences, University of Alberta

[Figure]

General comments:

I would best describe this paper as a reconnaissance of an important new exposure in the interior of Yakutia. The site is remarkable in its height and no doubt was a very difficult exposure to work. I have worked on similar sites in Alaska and so commend the authors on the amount of information and samples they were able to collect from the site. It is no small task and I appreciate their comments about the danger of some parts of the exposure.

The stated purpose of the paper, and I assume the choice of journal, is to reconstruct the climate of this interior continental site in central Siberia via cryolithological reconstruction. It does this generally through reconstruction of the depositional environments, and to some extent the record of active ice wedges and some plant macrofossil data. The cryofacies part is not well developed and I think, particularly for the origin of Unit IV this could be useful. On this theme were there no water isotope samples collected for the site? These would be useful for the origins of Unit IV. The choice of journal did not seem immediately obvious, but that being said, the largely lithological reconstructions in the paper, coupled with paleobotanical data, do represent proxy records that tell us about past climate for this region.

I read the paper with interest given the location of the site, but that being said I was a bit disappointed that the chronology did not work out better to allow a clearer identification of the MIS 3/2/1 units (via radiocarbon) coupled with the OSL dating for the lower purported MIS 5 units below. The radiocarbon dating is sparse and I am surprised of the uncertainty toward the top of the exposure, particularly given the presence of the Arctic ground squirrel midden. Only a single sample was dated from this midden, but an abundance of discrete macros must be available, no? And the suggestion that the AGS may have burrowed below the overlying 33ka data is unlikely- modern AGS's are well established in that they will only burrow to the depth of the active layer and they tend to only be present on sites with thicker active layers (up to 1m or so). This is well established in the North American literature. An additional age on this nest should

confirm if the 26ka date is accurate, with the implication of a problematic overlying age or reworked macrofossil from older deposits.

In terms of evaluating the radiocarbon, are there any QA standards from the radiocarbon lab that would allow us to know what background was? Were these small mass samples? Was there a mass-dependent background blank if these were small samples? At present it is hard to evaluate the contradictory radiocarbon results and I would assume given the sequential lab numbers the radiocarbon lab would be able to provide additional information that may help with understanding these inconsistencies- i.e. are these inconsistencies reflecting reworking of older macrofossils or reaching background with small mass samples or some other problem?

The stratigraphic reconstruction the authors present seems quite reasonable and they make a strong case through correlation to the coastal sites in their previous work. In that sense they have set up a useful stratigraphic framework for the site, and no doubt this will enable future work to focus on specific intervals- such as the last interglacial or the MIS3/2/1 interval.

The one question that I think is perhaps of most interest to a broad paleoclimate readership is the extent of thaw during the last interglaciation and whether this question is tractable at the Batagay site. It would appear to be. On p11 they discuss the Unit III layer and indicate a relatively uniform thickness of about 1m reaching up to 3.5m in ice wedge casts? It would be useful to see more documentation of this unit. Are there relict ice wedges below this unit to indicate the depth of permafrost thaw during that time (it would appear so from the descriptions and photos)? What is the nature of the cast fill/thaw unconformity? Were these casts sampled for macrofossils? Our own experience has been that many of the thermophilous taxa are preserved in cast fills (see Kuzmina et al., 2014 Quat. Int.) because of the accommodation space provided in the wedge. Is the infill waterlain/stratified perhaps indicating thermokarst ponds? Or was there relief and the potential for past intervals of retrogressive thaw slumping similar to today? My guess would be the latter. It was not clear to me the relations with

the underlying Unit IV that seemed to lack relict ice wedges, but they are present in the underlying Unit V. But does Unit IV include relict pore ice? It appears syngenetic and anoxic from Figure 6, but can you add more to this? What was the extent of the thaw of this unit with the unconformity of Unit III? What are the cryofacies/structures of this unit? Is there evidence of thaw and refreezing of this unit epigenetically? Water isotopes would seem a useful tool through this stratigraphic sequence.

Overall, however I do not see any of these comments as being fatal to the manuscript, and I don't expect that all of these questions are tractable from this first investigation of the site, but I do hope future work will be forthcoming. I think the paper is well organized and I recommend its publication in COPD with minor revision to the text for clarity and perhaps some large discussion around the paleoenvironmental significance of the exposure to some broader questions. The Batagay site is remarkable and I think will yield important insight into the ca. 150,000 years or so of earth history preserved at the site and I look forward to reading about it.

Specific comments:

1. Title- It is too long. I suggest Paleoenvironmental reconstruction of MIS 6-1 relict permafrost from the Batagay mega thaw slump in interior Siberia (or some similarly shortened version of the title). 2. Chronology: this is the weakest part of the paper and I'm sure a source of frustration to the authors for understanding the significance of the lithostratigraphic units. For the most part the authors have taken their coastal stratigraphy and applied their understanding to the main units at this site. I think this is entirely appropriate for a reconnaissance survey and no doubt will be followed up with future work to test these correlations through detailed independent chronology. 3. The cryo part is inconsistent. Looking through Table 1, there are cryofacies (or at least some ice descriptions only for a few units) and ultimately no photos of the cryostructures at the site. This would be particularly useful, if they exist for Units 3 and 4 and should be added to Table 1 and perhaps some of the descriptions of the units in the main text where they assist in the interpretations. 4. I'm surprised to not see any

water isotope data in this paper? Was there a special problem? If the stated purpose is a paleoclimate reconstruction, the water isotopes would be particularly useful for the last interglacial unit in particular and in understanding the origins of Unit IV and the likelihood for Unit V to be MIS 6. 5. Table 4- the authors present a correlation of the main units with MIS's and permafrost dynamics. This is largely based on their correlations to the coastal sites and so I think needs to be restated as more speculative to acknowledge the uncertainties in the dating at the site. It seems likely but it is a supposition. 6. Figure 3 the geological map does not add much in my opinion- I would suggest stating something about the geology in the intro/site description, but dropping this figure. 7. Figure 6- there are no descriptions for panels f or g- which I assume come from units 3 and 5?

---

## Editor Comment (EC2) · D.-D. Rousseau (Editor) · 23 Dec 2016

Dear authors , As the second reviewer recently posted his review, could please provide a general reply to this detailed report so that I could consider wheter of not I invite you to submit a revised version of your manuscript? All the very best, denis-didier Rousseau (Climate of the Past Co-Editor in Chief)

---

## Author Comment (AC2) · 4 Jan 2017

Dear Professor Froese,

we thank you for your time and efforts to review our manuscript and also for your recommendation to accept it after minor revision for *Climate of the Past*. Your helpful comments, suggestions, and examples from your extensive experience give us confidence to improve the manuscript.

[Figure]

We appreciate your suggestion to reflect the reconnaissance character of the manuscript in the title and the text.

The methods and aims of the project as well as the equipment available in the expedition were not supposed to conduct water isotope analyses. We agree that this would be of a high value in reconstructing the evolution of the sequence, and we will definitely heed this advice for future expeditions. But so far, some of the questions stated in the review cannot be answered with our data (especially addressed to Units IV and V).

We understand your concerns on the chronology provided by the radiocarbon dating results. Indeed, we sampled all the available material from the Arctic ground squirrel midden and double dated it. Unfortunately, the funding of our project doesn't allow for more detailed radiocarbon dating. We will contact the Poznan Radiocarbon Laboratory in order to specify the useful additional information you kindly mentioned that could help us with improving the paper.

Thank you for your questions on the Unit III, they will guide us on the way to significantly improve the manuscript. There are more general pictures from Unit III and we will include them in the documentation of Unit III. But, as it was mentioned in the text, it was possible to approach and sample this unit only at one site; we will check the field notes and photo documentation once more to make sure that all observed details are given in the text. We also sampled the casts for the macrofossils and will publish the palaeobotanical data of the profile in another paper.

We appreciate your comments and will implement them and our own ideas on enlargement the paleoenvironmental significance and for revision our manuscript.

---

## Author Response (AR1)

***Editor Decision: Reconsider after major revisions*** *(19 Jan 2017) by Dr. Denis-Didier Rousseau*
*Comments to the Author:*

*Dear authors,*
*We have now the two reviews and I would like to thank you for following my*
*recommendations in providing your comments to those reviews. The reviews, although*
*raising interesting but important topics, are quite positive and therefore I am sure, reading*
*carefully some of your comments that you could improve your manuscript accordingly. So I*
*am pleased to invite you to submit a revised version. Please follow once more Copernicus*
*process.*

*I stress you to follow the first reviewer about his concern about the paleoclimate issue and*
*possibly develop that part a bit more. Now, because also of those topics raised, I would like to*
*get another advice from the reviewers, the reason why I am selecting "major revisions" which*
*is the only way for me, through the Copernicus system, to get the reviewers' comments on*
*the revised paper.*

*Thank you once more for submitting your results to Climate of the Past and I look forward to*
*read your revised manuscript.*
*All the very best*
*denis-didier Rousseau*
*Climate of the Past co-editor in chief*

Dear Dr. Rousseau,

we would like to thank you, Dr. B. Gaglioti, and Prof. D.G. Froese for the reviews. We have
changed the manuscript according to your suggestions and with respect to helpful
comments and questions of both reviewers.
Following, we provide our point-by-point reply to the comments and outline the changes
made in the manuscript before resubmission.
The structure is as follows:

**Reviewers´ comments** (RC#1 for Dr. B. Gaglioti, RC#2 for Prof. D.G. Froese)
Authors reply (AR)
Changes (Page, Line)
Changes in the manuscript text

Major changes:
- We would like to change the title of the manuscript in order to underline the preliminary
surveying character of the work. We report on a permafrost exposure that was scarcely
studied before. We make attempts to correlate the sequence with coastal permafrost
sequences using mainly chronostratigraphical methods in order to trace the differences and
similarities. We do not try to hide information or, on the contrary, to state that the methods
we used are the most appropriate possible. We applied the methods that were feasible for
us under the given circumstances and with the available equipment in that particular
expedition. We also support the idea that there should be used a variety of other methods in
order to answer all the questions.
- Abstract is revised.
- We rearranged the Results section, according to the reviewers´ suggestions.
- The Discussion section is strengthened, according to the reviewers´ suggestions.
- Additional information on the radiocarbon dating is included in the Table 2.

- The figures with the overview map of all localities mentioned in the text (Figure 3), pictures of Unit III were added (Figure 8). The geological map of the region was dropped.

Additionally, we would like to draw your attention to a number of additional, minor changes that have been made to the manuscript. The changes have been highlighted on the attached version of the revised manuscript.

**Response to Comments**

RC#1*: Page 1 Line 14: Late Pleistocene should be capitalized.*
AR: Revised.
Page 1, Line 37.

RC#1*: Page 1 Line 18: '…sought climate record.' Should be reworded.*
AR: Removed.

RC#1*: Page 1 Line 19: 'close by the pole of cold' I am not sure this name is well known.*
20  AR: Revised.
Verkhoyansk instead of Pole of Cold.

RC#1*: Page 1 Line 25: '8° C colder than today' What is this quantified reconstruction of MIS 8 based on? Are they talking mean annual temperature? See below for more on this.*
25  AR: We didn't provide any quantitative reconstruction of MIS 8 but simply described the existence of a 'Middle Pleistocene Ice Complex'. The Middle Pleistocene comprises MIS 19 to MIS 6, thus also MIS 8, but we are not able to delimit the age of that Ice Complex more precisely nor are we able to provide any climate data from MIS 8.
As is described on page 13, line 9, YIC deposits formed during MIS 2-MIS 4 at a ground
30  temperature 8° C lower than today (Romanovskii et al. 2000b). Such conditions can therefore be assumed also for the aggradation of Ice Complexes older than MIS 4.
We wrote about ground temperature that is defined as the temperature of the (frozen) ground in the depth (distance from the ground surface downward) of zero annual temperature amplitude, where the ground is unaffected by seasonal temperature
35  fluctuations.

RC#1*: Page 1 Line 28: 'proves again' should be reworded.*
AR: Revised.
Page 1, Line 25.

RC#1*: Page 1 Line 30: 'In the Holocene cover….' I think the authors mean in the Holocene unit.*
AR: Removed.

45  RC#1*: Page 2 Lines 6-10: This final statement of the abstract is one side of an ongoing*

*controversy about the origin or processes that generate yedoma deposits. If the authors are going to interpret such deposits as being formed by nival and proluvial processes, then I think they should briefly describe the basis for this interpretation.*

AR: we would not like to discuss it in the abstract. This is discussed in the discussions section.

*RC#1: Page 2 Lines 22-25: This portion of the introduction describes the controversy of what geomorphic process is the cause of the yedoma ice complex. The authors are questioning whether aeolian deposition was primarily responsible because 'there existed a diversity of habitats, including aquatic'. I do not understand why the existence of aquatic habitats precludes the aeolian interpretation. Permafrost can perch the water table near the surface and this can create aquatic habitats in otherwise dry environments. I think the authors may want to describe better the basis for why they are questioning the aeolian hypothesis, and how this study can address this controversy. In addition, this controversy seems to be the main theme of the paper, and not in line with the Climates of the Past Discussions Journal*

AR: Revised. We are convinced that the discussion about Aeolian deposition certainly fits into the scope of the journal as aeolian deposition is exclusively climate driven.

Page 2, Lines 4-5.

… but the assumption that loess covered the whole area during the late Pleistocene contradicts cryolithological studies …

*RC#1: Page 3 Line 32: 'Globally greatest temperature gradient' should be reworded.*

AR: Revised.

Page 3, Line 11.

Meteorological observations recorded continuously since 1888 revealed the globally greatest temperature range at the Verkhoyansk weather station..

*RC#1: Page 4 Line 4: 'Accepted as the lowest temperature in the Northern Hemisphere' If there is a citation for this, then it should be called here.*

AR: Revised.

The citations are added.

Page 3, Line 14.

(Lydolph, 1985; Ivanova, 2006)

*RC#1: Page 4 Line 11: 'Resembling' should be 'Similar'*

AR: Revised.

Page 3, Line 26.

*RC#1: Page 5 Line 12: 'Possible reservoir effects as a result of the accidental use of freshwater aquatics…' This does not make sense. How did the authors know the macrofossils were aquatic? Did they identify them as such or did they infer this based on the ☐13C values. In addition, the authors should indicate what 14C calibration curve was used.*

AR: We identified the macrofossils, so that we know that none of them we of aquatic origin. Earlier in the text, we discussed that we used exclusively macrofossils that were identified as originating from terrestrial plants. We added the information on the calibration curve in the text.

Page 4, Lines 21-22, 26-27.

The calibration was made with the OxCal software (Bronk Ramsey, 2009) using the IntCal 2013.

RC#1*: Page 5 Line 16 to Page 6 Line 7: I am not an expert in OSL dating, but the methods described here seem to follow standard techniques in the literature.*

AR: We would like to keep the description of these techniques. The water content was not calculated in the field and in this case the accurate procedure of the age estimation should be reported.

RC#1*: Page 6 Line 10-16: It seems unnecessary to describe, in detail, how the thaw slump is positioned and behaves to start off the results section. This section does not seem to have much bearing on the main points of the paper. If this section needs to be included in the paper, then I suggest it go in the Study Site section. If the point of this is to say that the depths of different sections of the slump cannot be compared because some of them are not vertical, then this could be reduced to a few sentences.*

AR: We would like to keep this part as it is.
We feel confident that the detailed description is essential in the field observation and sampling part of the Results section. Additionally, these differences could be crucial for the discussion of the unconformities occurring along the exposure.

RC#1*: Page 6 Line 22: I assume these meter calls are being measured from the top of the slump. This should be specified here.*

AR: Revised.
Page 5, Line 28.

RC#1*: Page 9 Line 26: The authors think that the sedimentary transitions of the different units represent erosional unconformities. Do they see cut and fill or other features to back this up? I do not doubt this interpretation, but it would be useful to describe the reasoning behind this. In my opinion this interpretation can be in the Results.*

AR: We report on the transitions from unit to unit in the Results section of the manuscript. The description of each unit ends up with the features of the boundary to the next underlying unit, as specified in the Table 1 and in the text. We made sure that the information from the table is backed up by the text. We find appropriate to keep this short summary in the Discussion Section.
Page 5, Lines 31-32; Page 6, Line 16; Page 7, Lines 19-20.

RC#1*: Page 9 Line 31: I find the 300-year BP 14C age on plant material that is 1.15 m below the surface to be suspect. Is there loess deposition in this region today? How could 1.15 m of sediment accumulate in 300 years without incredible rates of productivity, a mass movement above the section, or high rates of loess deposition? Bluff-top sequences of loess in section often have reworked loess that blew onto the ground surface as the cutbank neared the site of the section. Is this a possibility? It would seem more likely that this date represents modern material from rooting or cryoturbation from the current vegetation mat.*

AR: The grain size distribution indicates that the main constituent of the sediment is not loess but sand, which admittedly doesn't excludes an aeolian (re-)deposition. In fact, there are sand dunes only a few km away near the Yana River floodplain (Page 11, Lines 27-32). No features of cryoturbation were observed but the horizon included roots of modern plants.. Cryoturbation is very unlikely since the ground is not wet enough for this process due to inclination and fast drainage. Aeolian transport does take place in the region nowadays.

Unfortunately, we do not have dating results of a higher resolution. Due to this reason we do not focus on Unit I in the manuscript, but briefly discuss this date.
Page 9, Lines 8-10.

RC#1*: Page 10 Lines 1-8: The dating results from this study do not necessarily indicate that the YIC was deposited continuously from 51-12 ka. This is because there are only a few 14C ages from this unit and they seem to be subject to reworking. Could an alternative view be that YIC accumulated only episodically or during a fraction of this time period because the plant remains were reworked or the deposits are too coarsely dated to infer continuous deposition.*

AR: The problem of potentially interrupted sedimentation was discussed in detail later in the discussion of the original manuscript on page 12, lines 13-20. The coarsely dated sequence does not allow us to claim that the YIC was deposited continuously from 51 ka to 12 ka BP. We referred to this problem on Page 10, Lines 36-37. To stress this we revised the text. Page 8, Lines 40-41; Page 10, Lines 36-37.

The coarse dating does not imply a continuous sedimentation during the last 51 ka, thus we cannot exclude interruptions in the sedimentation record.

RC#1*: Page 10 Lines 25-27: The authors say that the 14C age reversal could be due to a ground squirrel stashing food underground, which would bring younger C down below older C in section. The two dates are 26.2 and 33 ka. The younger date is 2.55 meters below the older one. Because ground squirrel food caches are limited by permafrost (they have not been observed to burrow into frozen ground), this would suggest that the active layer at this site was at least 2.55 meters deep. This does not seem plausible. The authors should discuss this further if they think it to be possible.*

RC#2*: The radiocarbon dating is sparse and I am surprised of the uncertainty toward the top of the exposure, particularly given the presence of the Arctic ground squirrel midden. Only a single sample was dated from this midden, but an abundance of discrete macros must be available, no? And the suggestion that the AGS may have burrowed below the overlying 33ka data is unlikely- modern AGS's are well established in that they will only burrow to the depth of the active layer and they tend to only be present on sites with thicker active layers (up to 1m or so). This is well established in the North American literature.*

*An additional age on this nest should confirm if the 26ka date is accurate, with the implication of a problematic overlying age or reworked macrofossil from older deposits.*

AR: We re-dated material from the AGS nest and obtained a confirmation of the first dating. The plants gathered by the AGS are now the most reliably dated material of the study. The assumption of plant material transport by ground squirrels is reasonable for depths up to 1 m below ground, which is an average depth for the permafrost table. The permafrost table as natural barrier for ground squirrel penetration can be even deeper, when the soil substrate is dry and coarse-grained as it is often the case for sandy deposits. Larionov (1943) reports on a ground squirrel nest found in Siberia in 2 m depth. Larionov, P. D.: Ecological survey of the yakutian longtail ground squirrel (Citellus eversmanni jacutensis Brandt), Zoological magazine, XXII, 4 :234—246, 1943.

This statement was an attempt of explanation the age reversal and, of course, it is speculative. Together with the fault tolerance of the radiocarbon dating, it might however help understanding the inversion. The substrate at the site is sandy and, during the lifetime

of the AGS, it was probably dry also due to the inclination at this slope. We consider however the eventuality that the overlying older age originates from redeposited material.
Page 9, Lines 15-24.

RC#1*: Page 11 Lines 1-5: The authors describe how the erosional unconformity probably corresponds to a thermal erosion event during the warm times of the Pleistocene Holocene transition. This may be true, but it should be acknowledged that the 14C dates that bracket this erosion event seem to span around ~25.7 ka. The Bolling Allerod and early Holocene warm interval were millennial-scale events. I think it should be acknowledged that this correspondence is highly speculative given the age control.*

AR: We assumed that there was an erosional event. We wanted to say that the (post-depositional) erosional event removed sediments that were deposited in the 25.7 ka before. According to Occam's razor, this is the best explanation. We revised the text and made it clear that the Bolling Allerod was an example of such event.
Page 9, Lines 30-32.

The Kargin intersdatial (MIS 3) was characterized by 3 warming phases in Yana-Indigirka lowland (Fradkina et al., 2005).

RC#1*: Page 11 Line 22: The authors say that the climatic conditions were insufficient for ice wedge growth, but climate is only part of driver for ice wedge growth. The type of depositional environment and grain size of Unit IV could also prevent ice wedges from forming or being preserved. The authors should rule out whether non-climatic factors contributed to the lack of ice wedges in this unit.*

AR: We revised the text to stress that the climatic conditions were not appropriate for Ice Complex formation (thick ice wedges as observed in Unit II). Thin ice wedges are preserved in Unit IV.
Page 10, Lines 6-9.

The lack of thick ice wedges or ice wedge casts indicates that the climate conditions during deposition of Unit IV were inappropriate for the formation of a pronounced Ice Complex directly below the last interglacial Unit III. Unit IV instead represents sediments that, in contrast to YIC deposits, consistently accumulated under uniform depositional environments.

RC#1*: Page 12 Line 1: The authors state that the presence of ice wedges in Unit V indicates that the mid-Pleistocene was characterized by extremely cold winters. This statement does not seem to be based on any dating, and relies on stratigraphic order. It should be acknowledged that just because the ice wedges are below the MIS 5 paleosol that this Unit V does not necessarily represent the Mid-Pleistocene. Similarly, the authors state that the ground ice in Unit V survived multiple interglacial warm times, but they only show that the ice survived MIS 5.*

AR: Due to absence of samples from Unit V, we provided a chronology based on stratigraphy and assert that climatic conditions cold enough for Ice Complex formation prevailed during the Middle Pleistocene. Similar observations of an ancient Ice Complex are available for the coastal Arctic and were dated with the 230Th/U method to MIS7.
The ground ice in Unit V still exists, so it is obvious that it survived several full interglacial warm stages including the Eemian and the Holocene as well.
We revised the text according to the Reviewers suggestions.
Page 10, Lines 22-27.

The finding of such ancient ice wedges demonstrates also that ice-rich permafrost survived at least two glacial-interglacial cycles (MIS 5 and MIS 1). Similar observations of Ice Complex deposits older than the last interglacial were made on Bol'shoy Lyakhovsky Island by Andreev et al. (2004) and Tumskoy (2012) and were dated by Schirrmeister et al. (2002) to MIS 7. On the basis of the stratigraphical position of this Ice Complex below Unit III, which is supposed to be deposited during the last interglacial, we assume that Unit V is older than MIS 5e, thus of mid-Pleistocene age.

RC#1: *Page 12 Line 25-30: The authors state that the only mechanisms for the deposition of >50% sand in Unit II come from proluvial, nival, or periglacial processes, but give no citation. The authors do not think that aeolian processes could be responsible for depositing this unit. In many aeolian settings, silt and sand can be deposited together depending on sediment availability, wind energy, and surface roughness. It is not uncommon to have sand sheet interbedded with loess deposition. A more detailed report on the sedimentary facies in the section could constrain whether aeolian processes are at play here.*

AR: We did not state that periglacial, proluvial, or nival processes are the only possible processes resulting in this GSD but listed them as the most probable ones according to our understanding. The discussion section of the Unit II consists of several paragraphs introducing the possible sedimentation processes one after another. We do not state that the sand deposition could not be a result of aeolian processes. The paragraph (Page 11, Lines 24-35) is dedicated exclusively to the aeolian transport.
We revised the text and inserted the citation.
Page 11, Line 19.

RC#1: *Page 13 Lines 13-15: A citation call would be useful to back up the interpretation that changes in magnetic susceptibility is a prerequisite for aeolian deposition here. I think there could be little change in MS under varying aeolian processes.*

AR: In most of the samples from Unit II (YIC) the magnetic susceptibility ranged between 17.1 and 27.1 SI. Only in the depth of about 32 m bgs the MS is higher (31.2-to 42.8 SI). But this is the lower boundary of the YIC, where possibly erosional processes occurred. This is not correlated with changes in the GSD signatures. In the loess literature significant changes in MS which were used for palaeoclimate interpretation are about one order higher than the differences we have measured from the Batagay profile (see e.g. Jimin Sun, Tungsheng Liu (2000). Multiple origins and interpretations of the magnetic susceptibility signal in Chinese wind-blown sediments Earth and Planetary Science Letters 180, 287-296.)
Again, we never denied that aeolian processes are involved in the genesis of the YIC. But it was always a polygenetic process.

RC#1: *Page 13 Lines 17-20: It is not clear to me why free-thaw action on quartz grains excludes the possibility of aeolian deposition here. Wouldn't freeze-thaw action be prevalent in this region regardless of the climate or depositional environment?*

AR: We did not state that free-thaw action excludes Aeolian deposition. We tried to stress that the high percentage of the silt fraction in the GSD of subunit IIb cannot be interpreted as an *exclusive* indicator of aeolian deposition, but free-thaw action could also have an input.
We revised the text to make our point clear.

Page 11, Lines 41-42.

The predominance of silt in the GSD might be a result of the combination of both processes, frost weathering and aeolian deposition.

RC#1: *Page 13 Lines 29-30: The authors call MIS 5a the last glacial period. This seems too similar to the common name for MIS 2, which is often called the last glacial period. I suggest another name.*

AR: We now omit the correlation with MIS and changed the expression.

Page 12, Lines 34-37.

RC#1: *Page 14 Line 12: The mean annual ground temperature is only partly driven by climate. Surface processes, like the thermal conductivity of different soils and the thickness of the insulating snow layer, should be discussed as these features were likely different in the past.*

AR: We totally support the reasoning in this comment. That is why we specified in the original text that other parameters apart from the mean annual ground temperature played a role: *"Ice wedge growth is not only influenced by climate but also by local factors such as ice content, grain size distribution, vegetation and snow depth"*, Page 14, Lines 9-10.

On the basis of modern ice-wedge formation and examples available from the literature (Kaplina, 1981; Plug and Werner, 2008), we are however able to assert that climate conditions were colder than present.

Page 15, Line 14-16.

**Abstract in General:**

RC#1: *In general, the abstract is too long. It should be cut in half to describe the main motivation, approach, and points of the study. The order of the abstract is counterintuitive to the study. First, the authors introduce the site, and units with some specific temperature reconstructions. Then the authors describe detailed methods that they used including the sampling interval. These methods shouldn't be in the abstract, and certainly should not come after the main points of the paper are described. Similarly, details about organic carbon magnetic susceptibility etc. do not need to be in the abstract if they are not contributing anything about the main points of the paper. I ask that the authors reconsider describing the Siberian lowlands as a maritime climate. Potential evapotranspiration exceeds precipitation in much of the Arctic. Perhaps the authors mean that the region is less continental today than it was during glacial intervals when this yedoma deposit formed. The lowlands are also described as maritime in the Introduction.*

AR: We revised the abstract according to the comments. Now it is shorter, the methods, details on the organic carbon etc. are removed. The Siberian lowland climate is now discussed in the Introduction and Study Site sections in the revised version.

Maritime is a relative term and refers to the distance to the sea. The coastal lowlands are close to the sea and, thus, more maritime than the inland sites. This is well recognizable when comparing climate, especially temperature, data from coastal and inland sites.

Page 1, Lines 11-34; Page 2, Lines 9-12.

**Introduction in General:**

RC#1: *The authors describe the climate of the Siberian lowlands as maritime and the study area in the Yana highlands as more continental. I suggest the authors include the mean climatic specifications to show how different the two regions are.*

*I also question whether these two sites were climatically different when they formed*

*during past glacial periods. Because eustatic sea level was much lower, and permanent sea ice more extensive the whole region would have been more continental, and the lowlands would have been included in this. Therefore, the authors should describe how much different these areas were in the past. The final few paragraphs in the introduction are better suited for the Study Site section as they describe the study site.*

AR: Continental climate is characterized by relatively low precipitation and a great seasonal (or in lower latitudes diurnal) temperature gradient forming under the influence of a large landmass and the great distance to the sea. These coastal sites were thus under continental climate influence during cold stages and under more maritime climate influence during warm stages. This is a difference to the Batagay site, which was always far away from a coast. These differences make the site and its comparison with coastal sites so interesting. We revised the text and highlighted these differences in the chapter Study site as well as included the names of the climates according to the Köppen climate classification. Also the mean climatic specifications and their values for coastal and inland zones are now in the text. The differences between coastal and inland sites in the past are described in the discussion part.

Page 2, Lines 9-12; Page 3, Lines 7-14, 16-20.

**Study Site in General:**

RC#1*: I suggest that the authors briefly describe the modern-day vegetation, and major geomorphic processes occurring in the region today aside from the slump.*

AR: A short overview of the modern vegetation is now included in the text.

Page 3, Lines 33-36.

**Results in General:**

RC#1*: In my opinion, it is not necessary to describe the angles of the bluff and sections at various depths. These are subject to change within a few days of being described and do not add much to the interpretation.*

AR: we would like to keep it as it gives an impression of the study area and explains why most of the outcrop was not accessible for sampling.

RC#1*: The results would read much better if this section were to be broken up into different units instead of different techniques. The authors could easily describe the lithology, chronology, organic geochemistry, paleobotany, etc of Unit I and then proceed to Unit II. This provides a narrative for what these units are composed of and when they were formed. I think this approach would also save significant space.*

AR: The Results section is revised. We applied the proposed way of presenting the data separately for each unit.

Page 5, Line 8 – Page 8, Line 15.

RC#1*: There is a distinct lack of information about sedimentary facies in this paper. The interpretation could be greatly aided by these results. Was there bedding or was each unit massive? What general attributes did these beds/laminations have? Was there fine rootlets embed in the sequence to suggest that the landscape was covered by vegetation? Was there any soil development? If so, what horizons / weathering is present?*

RC#2*:  The cryo part is inconsistent. Looking through Table 1, there are cryofacies (or at least some ice descriptions only for a few units) and ultimately no photos of the cryostructures at the site. This would be particularly useful, if they exist for Units 3 and 4 and should be added*

*to Table 1 and perhaps some of the descriptions of the units in the main text where they assist in the interpretations.*

AR: We revised the text and made sure that all the crucial data mentioned in the table 1 is presented now in the text and vice versa as well. We checked the field notes and updated the information where it was possible.

As specified in the first paragraph of the section Methods, the other units were not accessible and could not be approached at this season, when the cliff walls (where the other units are outcropped) were thawing and chunks of sediments were continuously falling down the up to 60m high wall. There seem to be some palaeosol horizons in unit IV but we could not approach and omitted a remote diagnosis. Unit III seems to contain another pronounced palaeosol, but this layer was also out of reach for more detailed description.

Page 5, Lines 30-32; Page 6, Lines 1-3; 14-16; Page 7, Lines 18-22; Page 8, Lines 10-12; Page 13, Line 42; Page 14, Line 20; Page 22, Lines 24, 25, 27, 33, 39; Page 23, Lines 1-2, 5-10.

RC#1*: The type of material that was 14C dated should be described in the text. 'Plant remains' should be specified to taxa. It should also be specified how many aquatic plants were dated from this section, but not reported in this study. The methods give the impression that some 14C dates were excluded, but which ones, where were they sampled, and what were the ages?*

AR: The information on the dated material is included in the Table 2 (Radiocarbon dating..). We revised the name of the table in order to avoid the impression that the results are only partly reported. The plant macrofossils were identified if possible and the information is given in the table. Unspecified "plant remains" refer to unidentified terrestrial taxa, e.g. twigs and scales from shrubs or dwarf shrubs. There were no aquatic plants found in the samples, hence, none of them were dated (Page 4, Lines 21-22 in new version). We were aware of the reservoir effect and carefully paid attention that no aquatics were used for radiocarbon dating.

Page 4, Line 21-22; Page 7, Lines 5-7; Page 23.

***Discussion in General:***

RC#1*: The authors are assuming that the three sub-units in Unit II represent different marine isotope stages (MIS 4-2). Radiocarbon dating does not back up this assertion. It only seems to be based on the fact that there are three units and three isotope stages occurring at roughly this time. Much of the discussion on the possible links between the MI stages and subunits in Unit II could be removed.*

*The apparent erosional unconformities can be included in the Results if there areavailable sedimentary features that indicate where they are. As of now, most of the interpretation is based on large differences or reversals of 14C ages. This may not be warranted if the 14C dates are reworked, which the authors describe as a possibility. I am skeptical that the current resolution of dating allows the statement that the Batagay sequence is in good agreement with global climate events over the last 125 ka. Mostly this section shows the landscape response to the last interglacial warm times, but there are a number of other climate events since the mid-Pleistocene that may or may not be represented here. It is difficult to say with the unconformities and current dating resolution.*

RC#2: *Table 4- the authors present a correlation of the main units with MIS's and permafrost dynamics. This is largely based on their correlations to the coastal sites and so I think needs to be restated as more speculative to acknowledge the uncertainties in the dating at the site. It seems likely but it is a supposition.*

AR: We share your concern on the dating resolution. We base our interpretations and assumptions on several other proxies (description and comparison of the sedimentary facies, plant macrofossils) to connect dated parts of the record to dating results available from this and other regional permafrost sequences. We are confident that our stratigraphical interpretation is reasonable. We revised the table name in order to avoid misleading. Page 24, Lines 12-14.

Table 4. Overview of permafrost dynamics recorded in the Batagay sequence in presumable correlation with global and regional climate histories. Due to the sparse dating resolution, the correlation is mainly based on the chronostratigraphical comparison of Batagay and lowland exposures..

RC#1*: The plant macrofossil identifications should be in the results.*
AR: Revised. The detailed macrofossil identification results are in the Results section, Unit III. Page 7, Lines 5-7.

RC#1*: In my opinion, it makes much more sense to describe the depositional setting of the section prior to the paleoclimate interpretation. The authors ignore the possible interpretation that much of the sediment was reworked from the nearby floodplain by aeolian processes and deposited into the uplands. Because many of these rivers have nival flow regimes, large areas of exposed sediment would have been available for aeolian transport. This has long been described as the mechanism to get loess into the uplands in many periglacial zones. I think it should be considered here as well.*
AR: We described it as one of the sequence-forming processes even in the original abstract (last sentence) and in the discussion section as well. Furthermore, we discuss the Aeolian transport at Page 11, lines 24-35, emphasizing that even nowadays the wind storms are often and transport large amount of material. During the Pleistocene, when the continentality was even higher than it is now, the wind velocities were likely higher as well and the input of Aeolian transport in sedimentation processes is undisputable. Page 11, lines 24-35.

RC#1*: The authors assert that proluvial and nival processes were at least partly responsible for the deposition of fine-grained material in the section. A description of the sedimentary facies could help back up this claim, but this is lacking.*
*I suggest that the authors outline the methods and references used for identifying different depositional environments. These should be based on modern-day analogues with pictures of the modern-day depositional processes and their facies compared with these corresponding facies in section. In general the final section of the Discussion makes several assertions about the origin of the sediment without such comparisons.*
AR: The implementation of this suggestion would require many years of fieldwork and is not the scope of that paper. We doubt the existence of 'modern analogs' of depositional processes resulting in Ice Complex formation. Ice Complex is a relic of the Pleistocene and formed under cold stage climate conditions. Such conditions don't exist today.

RC#1*: There are a number of other sections from around Siberia mentioned in the Discussion. It would be helpful to include the general locations of these places in the Map figures.*
AR: We included another map: Figure 3. This overview map with all permafrost exposures is now also mentioned in the text.
Page 2, Line 35; Page 27, Fig. 3.

***Figures in General:***

*RC#1: The dating results should be much clearer then they are.*

AR: The dating results obtained for our research interests cover the outcrop with focus on the most interesting depths (with the material rich in plant remains, ground squirrel nest). The results presented in the current paper are more of a reconnaissance character. Unfortunately, our project budget does not allow us fine dating throughout the outcrop. But this undoubtedly must be taken into account by the next scientists working in the area. We would take it as a good instruction for further works.

*RC#2: In terms of evaluating the radiocarbon, are there any QA standards from the radiocarbon lab that would allow us to know what background was? Were these small mass samples? Was there a mass-dependent background blank if these were small samples? At present it is hard to evaluate the contradictory radiocarbon results and I would assume given the sequential lab numbers the radiocarbon lab would be able to provide additional information that may help with understanding these inconsistencies ie. are these inconsistencies reflecting reworking of older macrofossils or reaching background with small mass samples or some other problem?*

AR: We requested additional information in the Laboratory and the lab kindly provided us with the details. All analyzed samples were relatively large, and carbon masses after combustion were bigger than 1 milligram, except of samples where they were ca. 0.8 and ca. 0.9 mgC, what is still acceptable. The levels of background measured in different runs, were very close to one another, indicating that modern contamination introduced during sample preparation is rather constant, and close to ca. 0.3 pMC. When determining 14C ages of samples of unknown age, the Lab conservatively assumes that uncertainty of background level is 1/3 of the measured background. The results of replicated $^{14}$C analyses (i.e. Poz-78878 and Poz-80390) are included in the Table 2; it clearly demonstrates that the unexpected $^{14}$C ages of these 2 samples weren't caused by laboratory errors. So most likely the macrofossils of the overlying sample were reworked. We find it appropriate to state the dating results as it gives the temporal frame of the outcrop formation that is crucial for further detailed investigations.

The columns with background pMC, C content, and $\delta^{13}$C values are added to the Table 2. Pages 23, 24.

*RC#2: The one question that I think is perhaps of most interest to a broad paleoclimate readership is the extent of thaw during the last interglaciation and whether this question is tractable at the Batagay site. It would appear to be. On p11 they discuss the Unit III layer and indicate a relatively uniform thickness of about 1m reaching up to 3.5m in ice wedge casts? It would be useful to see more documentation of this unit.*

AR:
It is difficult to say if this thick plant material was deposited in ice wedge casts or in other depressions such as trenches etc. These depressions were not regularly distributed as we would expect it from Ice wedge casts but they were rather isolated. Ice wedge casts would be indicators for the existence of a former Ice Complex immediately prior to the formation of the organic layer. In this case, the surplus water from thawing Ice Complex would presumably result in the formation of small ponds. We accurately studied the plant macrofossils incorporated in unit III layer and found no aquatic plant species. The species

composition rather indicates dry open forest. Unit IV underlying unit III doesn't look like degraded Ice Complex. We added general pictures of the outcrop with special focus on the Unit III in the figures.
Page 7, Line 2; Pages 32, Fig. 8.

*RC#2: Are there relict ice wedges below this unit III to indicate the depth of permafrost thaw during that time (it would appear so from the descriptions and photos)? What is the nature of the cast fill/thaw unconformity? Were these casts sampled for macrofossils? Our own experience has been that many of the thermophilous taxa are preserved in cast fills (see Kuzmina et al., 2014 Quat. Int.) because of the accommodation space provided in the wedge. Is the infill waterlain/stratified perhaps indicating thermokarst ponds? Or was there relief and the potential for past intervals of retrogressive thaw slumping similar to today? My guess would be the latter.*
AR: Yes, we sampled one cast for macrofossils (Unit III) and it indeed contains many of thermophilous taxa. The infill in the lowest part of the cast was rather stratified (Fig. 7f), but no aquatic plant taxa were identified. Most likely, ponds could not establish. This could be caused by the relief, as you mentioned. Instead, there established terrestrial vegetation and developed a thick organic layer.

*RC#2: It was not clear to me the relations with the underlying Unit IV that seemed to lack relict ice wedges, but they are present in the underlying Unit V. But does Unit IV include relict pore ice? It appears syngenetic and anoxic from Figure 6, but can you add more to this? What was the extent of the thaw of this unit with the unconformity of Unit III? What are the cryofacies/structures of this unit? Is there evidence of thaw and refreezing of this unit epigenetically? Water isotopes would seem a useful tool through this stratigraphic sequence.*
AR: Unfortunately, we could not approach to the ice cliff and unit IV was not accessible for closer examination or sampling. That's why, we cannot add anything. All information we have from unit IV of the ice cliff is obtainable from the photos. We could approach closest in an optical way using a telephoto lens.

*Specific comments:*
*RC#2: Title- It is too long. I suggest Paleoenvironmental reconstruction of MIS 6-1 relict permafrost from the Batagay mega thaw slump in interior Siberia (or some similarly shortened version of the title).*
AR: We revised the title.
Palaeoclimate characteristics in interior Siberia from MIS 6 to 2: first insights from the Batagay permafrost mega thaw slump in the Yana Highlands

*RC#2:  Chronology: this is the weakest part of the paper and I'm sure a source of frustration to the authors for understanding the significance of the lithostratigraphic units. For the most part the authors have taken their coastal stratigraphy and applied their understanding to the main units at this site. I think this is entirely appropriate for a reconnaissance survey and no doubt will be followed up with future work to test these correlations through detailed independent chronology.*
AR: This is true, indeed. We considered this study as a first reconnaissance survey as we had to assume that there is no lithostratigraphic description available yet. Since this is the precondition for further interpretation, we made it by ourselves.

*RC#2: I'm surprised to not see any water isotope data in this paper? Was there a special problem? If the stated purpose is a paleoclimate reconstruction, the water isotopes would be particularly useful for the last interglacial unit in particular and in understanding the origins of Unit IV and the likelihood for Unit V to be MIS 6.*

*RC#2: The cryofacies part is not well developed and I think, particularly for the origin of Unit IV this could be useful. On this theme were there no water isotope samples collected for the site? These would be useful for the origins of Unit IV.*

AR: The methods and aims of the project as well as the tools in the expedition were not

10 supposed to conduct the water isotopes analyses. We agree that this would be very helpful in interpreting the results of our analyses, and we will definitely use this advice for the next expedition. But so far, some of the questions stated in the review cannot be answered with our data (especially addressed to Units IV and V since they were hardly accessible if at all). The fieldwork in Batagay was the result of a short-term opportunity. We didn't even have

15 enough time to obtain the equipment necessary for all useful investigations nor did we have access to all units. We also did not have enough time and manpower for all works. It was really only a first reconnaissance. There is left a lot to do for the future!

*RC#2: Figure 3 the geological map does not add much in my opinion- I would suggest stating*

20 *something about the geology in the intro/site description, but dropping this figure.*

AR: Revised. The information is in the text now, the figure is dropped.

*RC#2: Figure 6- there are no descriptions for panels f or g- which I assume come from units 3 and 5?*

25 AR: This is a small misunderstanding caused by the layout – Figure 6 consists of panels a-e. The Figure 7 is composed of panels a-g. The descriptions are below the Figure, we rearranged the spacing to avoid this failure in future.

[revised manuscript text omitted]